# TEMPORAL LABEL SMOOTHING FOR EARLY PREDICTION OF ADVERSE EVENTS

## ABSTRACT

Models that can predict adverse events ahead of time with low false-alarm rates are critical to the acceptance of decision support systems in the medical community. This challenging machine learning task remains typically treated as simple binary classification, with few bespoke methods proposed to leverage temporal dependency across samples. We propose Temporal Label Smoothing (TLS), a novel learning strategy that modulates smoothing strength as a function of proximity to the event of interest. This regularization technique reduces model confidence at the class boundary, where the signal is often noisy or uninformative, thus allowing training to focus on clinically informative data points away from this boundary region. From a theoretical perspective, we also show that our method can be framed as an extension of multi-horizon prediction, a learning heuristic proposed in other early prediction work. TLS empirically matches or outperforms all competitor methods on various early prediction benchmark tasks. In particular, our approach significantly improves performance on clinically-relevant metrics such as event recall at low false-alarm rates.

## 1 INTRODUCTION

Early prediction of adverse events is key to safety-critical operations such as clinical care [1] or environmental monitoring [2]. In particular, adverse event prediction is highly relevant to clinical decision-making, as the deployment of in-patient risk stratification models can significantly improve patient outcomes and facilitate resource planning [1]. For instance, the National Early Warning Score (NEWS), a simple rule-based model predicting acute deterioration in critical care units, has been demonstrated to reduce in-patient mortality [3; 4].

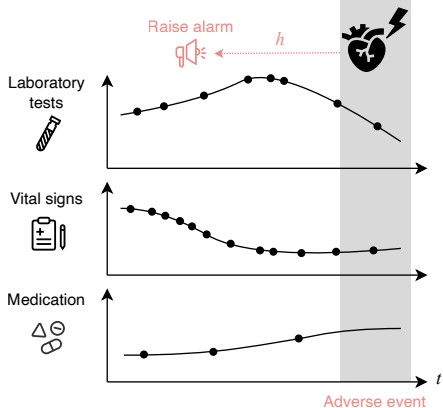

Figure 1: **Early prediction task.**

Deteriorating patient signals are often identified by mining large quantities of existing medical data and associated patient outcomes, which has sparked a growing interest in machine learning and medical literature. Applications of such adverse event prediction models include alarm systems for delirium [5], septic shock [6], as well as circulatory or kidney failure in the intensive care unit (ICU) [7; 8].

Adverse event prediction remains a challenging modeling task requiring specific technical solutions. Recent years have seen the development of deep learning architectures for electronic health records (EHR), which help tackle the high dimensionality, irregular sampling, and informative missingness patterns in patient covariates [6; 9; 10; 8]. Still, adverse clinical events are often noisy, infrequent, and, as illustrated in Figure 1, must be predicted with enough anticipation to allow for appropriate physician response – yet early prediction remains largely considered a simple binary classification task [7; 11; 9; 8].

As a result, current decision support models often suffer from high false positive prediction rates, with associated risks of alarm fatigue and thus limited physician engagement [12; 13; 1]. As highlighted in Figure 2a, the traditional cross-entropy objective results in the highest error rates near the class

boundary, corresponding to the prediction horizon before the event. Data in this boundary region dominates the loss but may not be clinically discriminative of patient deterioration patterns. Motivated by this observation, we propose Temporal Label Smoothing (TLS), a novel regularization strategy making label smoothing [14] time-dependent to better match prediction uncertainty patterns over time. As visualized in Figure 2b, our method is designed to reduce model confidence with stronger smoothing at the class boundary, allowing training to focus on more clinically informative data points away from this noisily labeled region.

**Contributions.** The contributions of our work are threefold: (i) In Section 3.2, we introduce a novel label smoothing method[1], which leverages the temporal structure of early prediction tasks to focus training and model confidence on areas with a stronger predictive signal. (ii) In Section 5, we show that our approach improves prediction performance over previously proposed objectives, particularly for clinically relevant criteria. (iii) In Section 3.3, we bridge the gap between prior work on multi-horizon prediction (MHP) [8] and label smoothing [14] by showing the former is equivalent to a special case of TLS under reasonable assumptions that we verify empirically.

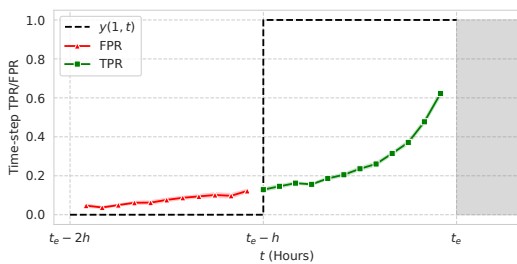 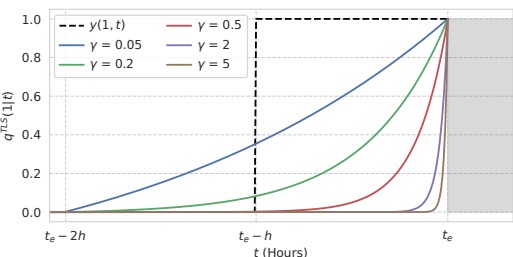

(a) Timestep performance of regular cross-entropy training for decompensation on MIMIC-III.

(b) Comparison of temporally-smoothed and ground-truth labels.

Figure 2: **Illustration of temporal label smoothing** for early prediction of adverse events. Predictions are carried out over a horizon $h$ and $t_e$ is the time of the next event, shaded in grey. True labels in black. (a) Model confusion is highest near the label boundary $t_e - h$ (maximum false positive, FPR, and minimum true positive rates, TPR), while performance is best close to event occurrence ($t_e$) and away from it ($t_e - 2h$). This motivates greater smoothing near $t_e - h$. Metrics are computed over four-hour bins based on a 50% precision threshold. (b) $\gamma$ controls the smoothing strength of surrogate labels $q^{TLS}$.

## 2 RELATED WORK

Recent years have seen the development of custom machine learning methods to predict expected patient evolution and support clinical decision-making [15; 16; 17; 7]. Amongst these, early prediction of adverse clinical events is a particularly complex task due to their typically rare occurrence and noisy label definition, which induces challenging, highly imbalanced datasets for model training [8]. As a result, prediction systems often suffer from high false-alarm rates with limited usefulness in the clinical context [1]. Prior works on early event prediction have adopted various approaches to tackle this issue, which we compare in Table 1 and formalize in Appendix A.4. We also discuss similarities and distinctions between our task and the frameworks of early time-series classification and survival analysis [18] in Appendix A.3.

**Learning objectives for imbalanced datasets.** Class imbalance is often addressed through loss reweighting techniques. Static class reweighting was used for sepsis or circulatory failure prediction [17; 7] through a balanced cross-entropy, which assigns a higher weight to samples from the minority class [19]. Still, performance improvements with this objective remain limited on highly imbalanced prediction tasks [20]. In contrast, dynamic reweighting methods such as focal loss and extensions [21; 22] induce a learning bias towards samples with high model uncertainty, typically harder to classify. This approach can improve the prediction of disease progression from imbalanced datasets [23] but does not consider patterns of sample informativeness over time.

---

[1]All code is made publicly available at `https://anonymous.4open.science/r/tls/`.

Table 1: **Related work.** Comparison with different training objectives for binary early prediction tasks. $y \in \{0, 1\}$ corresponds to a sample's true label at time $t$ and $\hat{y} \in [0, 1]$ to the model's prediction. Additional details and respective advantages of each work are further discussed in Appendix A.4.

| Related work | Temporal inductive bias | Computationally scalable | Impacts sample optimum | Loss for class $c \in \{0, 1\}$ |
|---|:---:|:---:|:---:|:---:|
| Cross-entropy loss [11; 7] | ✗ | ✓ | ✗ | $\delta_{y=c} \log(\hat{y})$ |
| Balanced cross-entropy loss [19] | ✗ | ✓ | ✗ | $\omega_y \delta_{y=c} \log(\hat{y})$ |
| Focal loss [21] | ✗ | ✓ | ✗ | $\omega_y (1 - \hat{y})^\zeta \delta_{y=c} \log(\hat{y})$ |
| Label smoothing [14] | ✗ | ✓ | ✓ | $q^{LS}(c|y) \log(\hat{y})$ |
| Multi-horizon prediction [8] | ✓ | ✗ | ✓ | $\sum_h y^h \log(\hat{y}^h)$ |
| **Temporal label smoothing** | ✓ | ✓ | ✓ | $q^{TLS}(c|y, t) \log(\hat{y})$ |

**Multi-horizon prediction.** In contrast, other early prediction models learn to leverage temporal trends in the data by outputting event predictions over several horizons [8; 24; 25]. This training heuristic improves prediction performance on the horizon of interest but scales poorly with the number of output horizons. In Section 3.3, we highlight that TLS can induce a similar temporal bias in learning while overcoming scalability limitations.

**Label smoothing.** For greater generalization of models applied to heterogeneous real-world data, another well-known training strategy is to avoid model overconfidence through label smoothing [14]. This regularization technique improves both the calibration of deep learning models [26] and their performance under noisy labeling [27; 26]. Still, despite extensions including novel prior distributions over classes [28] or modifications to the objective itself [29; 30], label smoothing remains designed for classification problems with i.i.d. samples, ill-adapted to the time-dependent nature of our data. To the best of our knowledge, we are the first work to explore adding a temporal dependence to label smoothing and empirically demonstrate the added value of this approach.

Whereas reweighted loss functions only bias learning towards minority or uncertain data points, multi-horizon prediction and label smoothing approaches alter the individual sample optimum. As a consequence, these approaches avoid model overconfidence and are thus more robust to noisy labeling [27]. In this work, we propose to combine the respective advantages of these established methods in a *novel way* to improve the early prediction of adverse events.

## 3 METHOD

We first formalize the problem of early adverse event prediction and introduce temporal label smoothing. We then highlight how MHP can be framed as a special case of TLS.

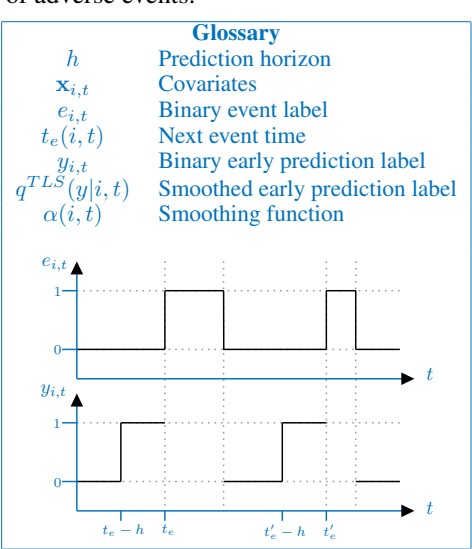

**Glossary**

| | |
|---|---|
| $h$ | Prediction horizon |
| $\mathbf{x}_{i,t}$ | Covariates |
| $e_{i,t}$ | Binary event label |
| $t_e(i, t)$ | Next event time |
| $y_{i,t}$ | Binary early prediction label |
| $q^{TLS}(y|i, t)$ | Smoothed early prediction label |
| $\alpha(i, t)$ | Smoothing function |

### 3.1 PROBLEM FORMALISM

We assume access to a dataset of $N$ patient stays. These consist of irregular time series of high-dimensional patient covariates $\mathbf{X}_{i,t} = [\mathbf{x}_{i,0}, \ldots, \mathbf{x}_{i,t}]$ and binary event labels $e_{i,t}$ encoding whether a patient of index $i$ is undergoing an adverse event of interest at time $t$. For each patient, we thus have a sequence $\{(\mathbf{x}_{i,1}, e_{i,1}), \ldots, (\mathbf{x}_{i,T_i}, e_{i,T_i})\}$ of length $T_i$.

Our early prediction task consists of modeling a binary target variable $y_{i,t}$, positive if the event occurs within a given prediction horizon $h$. For labelling purposes, we define the next event time for each time point, $t_e(i, t) = \arg\min_{\tau:\tau \geq t}\{e_{i,\tau} : e_{i,\tau} = 1\}$. If patient $i$ never undergoes any event, we set $t_e(i, t) = +\infty$. Thus, we have: $y_{i,t} = \mathbb{1}[t_e - h < t < t_e]$. As our task focuses specifically on *early* modeling for clinical relevance, no prediction is carried out if the patient is currently undergoing the event. Then, as for any binary learning problem, we define a model $f$ parameterized by $\theta$ with $\hat{y}_{i,t} = f_\theta(\mathbf{X}_{i,t}) = p_\theta(y_{i,t} = 1)$. We denote the optimal set of parameters minimizing the objective function as $\theta^*$, giving $y_{i,t}^* = f_{\theta^*}(\mathbf{X}_{i,t})$.

**Temporal structure.** An important distinction must be made with the classification tasks typically addressed with label smoothing. In adverse event prediction, data is not independent and identically distributed (i.i.d.) as each sample $x_{i,t}$ depends on a timestep $t$ and a patient stay indexed as $i$. Contiguous samples within a common stay are thus timely dependent:

$$p(y_{i,t+d} = 1) \geq p(y_{i,t} = 1) \quad \forall\, d : 0 \leq d < t_e(i,t) - t \tag{1}$$

Treating data as i.i.d, as is commonly done in early prediction works [7; 11], does not account for the increase in signal strength as the prediction time is approaching the event. Our goal is to leverage this structure in our data to *focus training on relevant timesteps* and help address issues of noisy label boundaries and class imbalance, which are inherent to our choice of real-world medical datasets.

### 3.2 TEMPORAL LABEL SMOOTHING

As introduced by Szegedy et al. [14], label smoothing consists of substituting the original label distribution, $\delta_{y_i=c}$ for class $c$, with a smooth version $q^{LS}(c|y_i)$ in the cross-entropy objective $L_i = L^{CE}(y_i, \hat{y}_i)$. For binary tasks, label smoothing becomes a linear interpolation:

$$q^{LS}(1|y_i) = (1 - \alpha)y_i + \alpha(1 - y_i) \tag{2}$$

where parameter $\alpha$ controls the smoothing strength.

By shifting the minimum of the objective function away from $y_i^* = y_i$ towards $y_i^* = q^{LS}$, label smoothing prevents models from becoming overconfident during training. This approach should therefore help improve the robustness of early prediction models against the inherently noisy nature of the task [27] but does not account for the time dependency between samples of a given stay. For this purpose, we propose temporal label smoothing, an approach to modulate smoothing based on time $t$ to infuse this prior knowledge into the training objective. We define the corresponding surrogate distribution similarly to label smoothing:

$$q^{TLS}(1|i,t) = 1 - \alpha(i,t) \tag{3}$$

For early prediction of events, to enforce the temporal inductive bias in Equation 1, we parametrize $\alpha(i,t)$ as a monotonously decreasing function of $t \in [0, t_e(i,t)]$. In practice, as illustrated in Figure 3a, this increases smoothing strength around the label boundary $t = t_e - h$, reducing prediction certainty in this region prone to high error rates, as shown in Figure 2a.

**Smoothing parametrizations.** We propose various temporal smoothing parametrizations for $\alpha(i,t)$ in Appendix A.2. Experimental results suggest that an exponential parametrization, defined as follows, performs best on considered tasks. Corresponding smoothed labels $q^{exp}(1|i,t)$ can be visualized in Figure 2b.

$$\alpha^{exp}(i,t) = \begin{cases} 1 - e^{-\gamma(t_e(i,t)-t-d)} - A & \text{if } h_{min} < t_e(i,t) - t < h_{max} \\ 0 & \text{if } t_e(i,t) - t \leq h_{min} \\ 1 & \text{if } t_e(i,t) - t \geq h_{max} \end{cases} \tag{4}$$

Parameters $h_{min}$ and $h_{max}$ define the time range over which we apply smoothing, namely $[t_e - h_{max}, t_e - h_{min}]$. Under this constraint, parameters $\{d, A\}$ are defined to enforce $\alpha(i,t)$ to be continuous at boundary points (see Appendix A.2). Finally, $\gamma$ controls the smoothing strength at a given time.

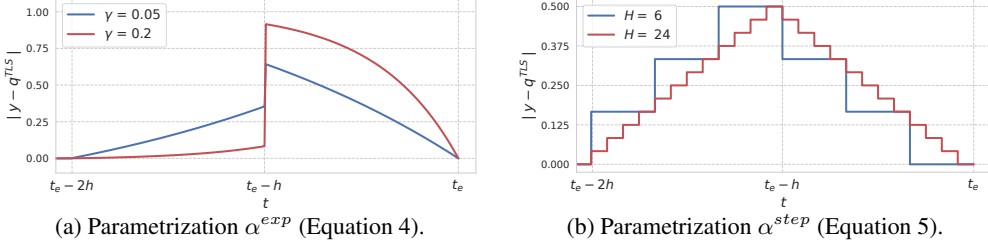

(a) Parametrization $\alpha^{exp}$ (Equation 4).    (b) Parametrization $\alpha^{step}$ (Equation 5).

Figure 3: **Label smoothing strength over time** under different parametrizations, with $(h_{min}, h_{max}) = (0, 2h)$. Note that $|y - q^{TLS}|$ corresponds to the difference in optimum $y^*$ between the TLS objective and cross-entropy. Smoothing function $\alpha^{step}$ is equivalent to multi-horizon prediction with a unique output.

### 3.3 Link with multi-horizon prediction

As motivated above, temporal label smoothing adapts the contribution of each sample to reflect prior knowledge about the temporal structure of event prediction labels. In this section, we find that MHP leverages the same information in Equation 1 to teach the model to predict events over multiple horizons/ [8]. Under simplifying assumptions justified empirically in Section 5.2, we show that this approach can be seen as a special case of temporal label smoothing with a 'staircase' parametrization.

In this framework, the unique label $y_{i,t}$ associated with patient covariates $\mathbf{X}_{i,t}$, for an horizon of interest $h$, is replaced by a vector $\mathbf{y}_{i,t} = [y_{i,t}^{h_1}, \ldots, y_{i,t}^{h}, \ldots, y_{i,t}^{h_H}]$ corresponding to $H$ distinct horizons. The prediction model is thus adapted to output $\hat{\mathbf{y}}_{i,t} = [\hat{y}_{i,t}^{h_1}, \ldots, \hat{y}_{i,t}^{h}, \ldots, \hat{y}_{i,t}^{h_H}]$. For temporal consistency between samples, Tomašev et al. [8] enforce predictions to be monotonically increasing over time, such that $h_u \leq h_v \implies \hat{y}_{i,t}^{h_u} \geq \hat{y}_{i,t}^{h_v}$. With these additional components, the training objective for patient $i$ becomes $L_i^{MHP} = -\frac{1}{H} \sum_{k=1}^{H} y_{i,t}^{h_k} \log(\hat{y}_{i,t}^{h_k}) + (1 - y_{i,t}^{h_k}) \log(1 - \hat{y}_{i,t}^{h_k})$.

**Proposition 1.** *Under the assumption that model outputs $\{\hat{y}_{i,t}^{h_k}\}_k$ are **equal** for all $\{h_k\}_k$ (rather than monotonically increasing), MHP is equivalent to temporal label smoothing parameterized with $\alpha^{step}(i, t)$. This function, illustrated in Figure 3b, is defined as the following sequence of step functions in time:*

$$\alpha^{step}(i, t) = \begin{cases} \frac{k}{H} & \text{if} \quad h_k \leq t_e(i, t) - t < h_{k+1} \quad \forall k \leq H - 1 \\ 0 & \text{if} \quad t_e(i, t) - t \leq h_1 \\ 1 & \text{if} \quad t_e(i, t) - t > h_H \end{cases} \tag{5}$$

*Proof.* See Appendix A.1. □

Proposition 1 frames MHP as a special case of TLS with step-function parametrization. We empirically justify the equal-output assumption through an ablation study in Section 5.2.

## 4 Experimental setup

### 4.1 Early prediction tasks

We demonstrate the effectiveness of our method on three clinical early prediction tasks with different characteristics, to understand its added value in each case. All tasks, established in existing literature and published benchmarks, deal with electronic health records from the ICU, where early prediction of organ failure or acute deterioration is critical to patient management [1].

Our work is first benchmarked on the prediction of acute *circulatory failure* and mild *respiratory failure* within the next $h = 12$ hours. These tasks are part of HiRID-ICU-Benchmark (HiB) [20], built on the publicly available HiRID dataset [7]. The dataset contains high-resolution observations of over 33,000 ICU admissions. Our third evaluation task consists of early prediction of patient mortality, or *decompensation*, within a horizon of $h = 24$ hours. Although less clinically relevant, this task has been widely studied in the machine learning literature [31]. Defined in the MIMIC-III Benchmark (M3B) [32], this task originates from the widely used MIMIC-III dataset [33], counting approximately 40,000 patient stays.

All three clinical events are labeled following internationally accepted criteria as in Harutyunyan et al. [32] and Yèche et al. [20]. Positive label prevalence is 4.3%, 38.6%, and 2.1% of timepoints for circulatory, respiratory failure, and decompensation prediction respectively – with rarer events associated with more severe states, in this instance. Further details on task definition and data pre-processing are provided in Appendix B.

**Signal deterioration over time.** As visualized in Figure 4, all tasks show a reduction in recall between event time $t_e$ and prediction horizon $t_e -$

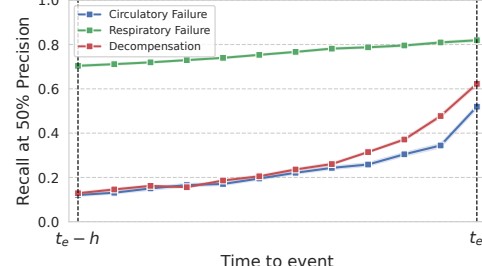

Figure 4: **Comparison of naive performance as a function of time** on all tasks, using a cross-entropy objective. Events should be predicted at a horizon $h$ from an event at time $t_e$. Model performance is reported at increments $\frac{h}{12}$.

$h$, suggesting a weakening in the discriminative signal associated with events and an increase in noise close to the label boundary. Whereas this performance decay is strong for the circulatory failure and decompensation tasks, respiratory failure shows a more consistent recall over time. From this observation, we expect that temporal label smoothing should improve performance on circulatory failure and decompensation prediction to a greater extent than on respiratory failure.

## 4.2 BENCHMARKING STRATEGY

**Baselines.** We quantify the added value of our method by comparing its performance to alternative learning approaches used for early event prediction, discussed in Section 2. Our first baselines consist of *balanced cross-entropy* [19] and *focal loss* [21], popular sample reweighting methods for imbalanced tasks. We also implement *multi-horizon prediction* as a multi-output model trained to predict event occurrence over different horizons between $0$ and $2h$. Note that for a fair comparison, we set $(h_{min}, h_{max}) = (0, 2h)$ in TLS. As in Tomašev et al. [8], a cumulative distribution function layer on logits enforces the monotonicity of predictions (Eq. 1). Finally, we also compare our method to conventional *label smoothing* [14] to confirm that a temporal dependency does improve performance.

**Hyperparameter tuning.** Hyperparameters introduced by our method, such as strength term $\gamma$ in smoothing parametrization $\alpha^{exp}$ (Equation 4), are optimized through grid searches on the validation set. The same approach is adopted for hyperparameters specific to each baseline, as shown in Figure 5.

**Architecture choice.** As our method and baselines are model-agnostic and only vary in terms of optimization objective, a unique model architecture is used for each task, selected through a random search on cross-entropy validation performance. Following a published benchmark on the HiRID dataset [20], we use a GRU [34] and transformer [35] architecture for the circulatory and respiratory failure tasks respectively. For decompensation prediction, transformers outperform the LSTM-based models [36] originally proposed in the M3B benchmark [32], and are thus used in our work. As recommended by Tomašev et al. [8], we apply $l_1$-regularization to input embedding layers, which improves performance on all tasks. Further implementation details are provided in Appendix C.

## 4.3 EVALUATION METRICS

To account for the imbalanced nature of clinical early prediction tasks, model performance is often reported through the area under the receiver operating characteristic curve (AUROC). Although this widely-used metric can be informative for moderate imbalances, the area under the precision-recall curve (AUPRC) provides more insight for our tasks: under a low prevalence of positive samples, precision is more sensitive to false alarms than specificity [37]. Still, "area under the curve" metrics can be poorly representative of clinical usefulness, as improvements in low precision regions can dominate such global metrics but remain incompatible with the low false alarm rates required for clinical deployment. Thus, to better assess model performance in this context, we also measure performance at a clinically motivated operating point through recall at 50% precision [24].

In addition to *timestep-level* metrics, which measure prediction performance at each data point, we also evaluate models in an *event-based* approach. Following Tomašev et al. [8]'s definition, an event prediction is positive if the model outputs a positive prediction at any time over the $h$ hours before the event. The threshold defining a positive prediction is chosen based on a precision lower-bound: in practice, we use a 50% stepwise precision criterion. This allows us to measure the event recall of our approach in comparison to published baselines. Unless stated otherwise, we always report mean performance with 95% confidence intervals computed over ten training runs.

## 5 RESULTS

### 5.1 PREDICTION PERFORMANCE

Overall, our results highlight that TLS improves performance over other approaches proposed to address the challenges of early clinical prediction. In Table 2, we find TLS to outperform other baselines across all metrics for both circulatory failure and decompensation. Despite over-lapping confidence intervals between multi-horizon and TLS on decompensation due to indi-

Table 2: **Timestep-level performance on different early prediction tasks.** Recall is reported at a 50% precision. Circulatory and respiratory failure are predicted on the HiB dataset, decompensation on M3B. In **bold**, we highlight best-performing methods with statistically significant $p$-values ($< 0.05$) under paired Student's t-tests [38]. As expected, performance gains on respiratory failure prediction are not significant on these metrics.

| Task | Circulatory Failure | | Decompensation | | Respiratory Failure | |
|---|---|---|---|---|---|---|
| Method | AUPRC | Recall | AUPRC | Recall | AUPRC | Recall |
| Cross-entropy | $39.1 \pm 0.4$ | $29.3 \pm 0.9$ | $34.5 \pm 0.4$ | $28.2 \pm 0.5$ | $60.5 \pm 0.2$ | $77.3 \pm 0.5$ |
| Label Smoothing [14] | $39.3 \pm 0.4$ | $29.9 \pm 0.8$ | $33.9 \pm 0.3$ | $27.7 \pm 0.5$ | $60.1 \pm 0.2$ | $76.6 \pm 0.5$ |
| Multi-horizon [8] | $39.6 \pm 0.5$ | $30.3 \pm 1.0$ | $34.9 \pm 0.3$ | $28.6 \pm 0.5$ | $60.3 \pm 0.1$ | $76.6 \pm 0.5$ |
| **Temporal Label Smoothing** | $\mathbf{40.6 \pm 0.3}$ | $\mathbf{32.3 \pm 0.7}$ | $\mathbf{35.5 \pm 0.3}$ | $\mathbf{29.3 \pm 0.4}$ | $60.4 \pm 0.2$ | $77.0 \pm 0.3$ |
| $p$-value ($H_0 : MHP \geq TLS$) | $\mathbf{0.00}^2$ | $\mathbf{0.00}^2$ | $\mathbf{0.00}^2$ | $\mathbf{0.02}$ | $0.15$ | $0.14$ |

vidual training run variability, we can reject the null hypothesis that MHP has a higher performance to our method ($p$-values $<$ 0.05), which supports the alternative hypothesis that TLS achieves superior results. Full precision-recall curves are given in Figures 6a and 14. This validates the experimental hypothesis proposed in Section 4.1, with, as expected, more limited improvements on respiratory failure – a result analyzed in greater detail in Section 5.3.

In contrast, as illustrated in Figure 5, loss reweighting methods designed to tackle class imbalance were found to reduce performance on all tasks over traditional cross-entropy. For weighted cross-entropy, we attribute it to the increase in false alarms resulting from the drive to improve recall. It further reduces the low precision of all models, thus negatively affecting the AUPRC (as visualized in Appendix D.5). On the other hand, focal loss down-weighs confident samples in training, constraining the model to focus on samples with uncertain predictions. In the context of noisy labeling, as is the case close to our class boundary, data points with ambiguous signals cannot be correctly predicted and thus dominate the loss, impeding improvements in other regions of input space. We analyze model performance over time in Section 5.2 to further support this hypothesis.

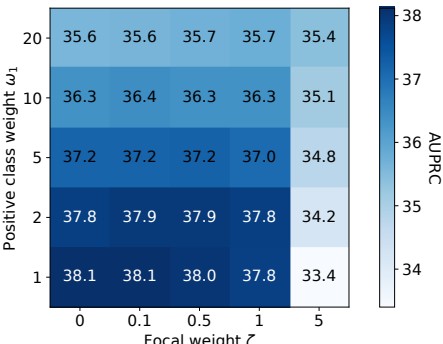

Figure 5: **Performance loss with class reweighting methods**, on the validation set for circulatory failure prediction. Weighted cross-entropy corresponds to $\zeta = 0$.

**Clinically-relevant performance.** We also compare the full precision-recall curve of models trained with these different objectives in Figure 6a – note that we obtain comparable results for decompensation prediction in Appendix D.2. In addition to visually confirming the numerical results in Table 2, we find that our training objective affords particular performance improvements in the clinically-relevant region corresponding with low false-alarm rates (precision greater than 50%)[1].

**Event-based analysis.** Finally, as highlighted in Figure 6b, TLS improves performance in terms of predicting overall adverse event *episodes* throughout a stay on all prediction tasks. This suggests that performance improvements at the timestep level affect a large number of events and translate to better event detection. Indeed, we demonstrate in Section 5.2 that TLS affords larger performance gains close to the event time, thus leading to a better recall of imminent events. We obtain similar conclusions for both other tasks (see Appendix D.1). For circulatory failure, temporal label smoothing is able to predict 7.4% more events than the closest baseline (multi-horizon prediction): this corresponds to reducing the number of missed events in the test set by a factor of 2, from 303 to 152 out of 2045 events on average. Within these events not captured by MHP, TLS predicts them on average 104 minutes before their occurrence, giving clinicians sufficient time to take action and avoid patient degradation.

---

[2]While some methods have overlapping confidence intervals on circulatory failure and decompensation prediction, TLS remains superior on each training run, giving $p$-values of 0.

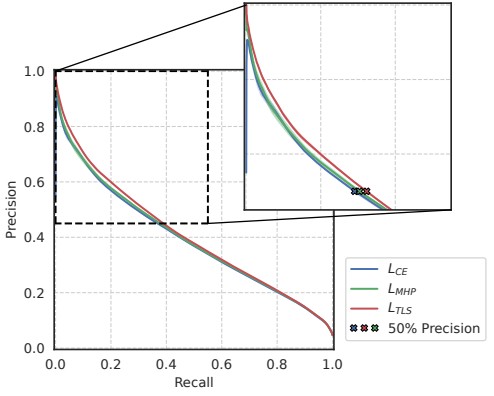
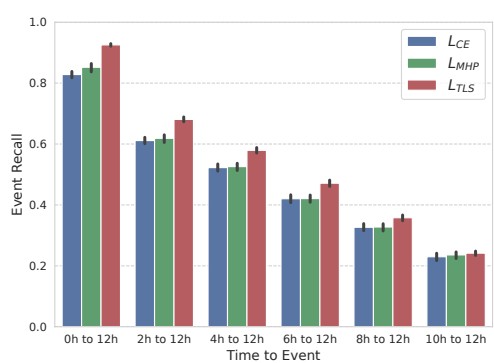

(a) Precision-recall curve. Inset shows the clinically-applicable region with precision > 50%.

(b) Event-level performance for a 50% timestep-level precision threshold.

Figure 6: **Clinically-oriented performance analysis** of different training objectives on circulatory failure prediction. See Appendix D for results on other tasks.

## 5.2 ILLUSTRATIVE INSIGHTS

We propose ablations and analyses to build intuition around our proposed method. In particular, we aim to highlight how temporal smoothing works and why it outperforms other training approaches for early prediction tasks.

**Performance over time.** In Figure 7, we compare the performance difference between our method, TLS, and the regular cross-entropy objective over time – previously studied in Figure 2a. We perform the same analysis in Appendix D for other tasks. As expected, the prediction model trained with TLS is less competitive where label smoothing is strongest, near $t_e - h$, but this performance loss remains minor even with significant smoothing. This result validates our hypothesis that the signal is too noisy in the boundary region for any model to recover the original label distribution. In contrast, away from the label boundary, TLS results in a significant increase in true positive and negative rates. From a clinical perspective, errors made in the boundary region are less critical, as they result in the latest false positives or earliest false negatives. Consequently, TLS not only improves global event prediction performance but allows these gains to occur at more critical times for clinicians.

**Empirical comparison to multi-horizon prediction.** In our theoretical discussion in Section 3.3, we demonstrated how MHP is a restriction of label smoothing with a step function $\alpha^{step}(i, t)$. This claim relies on the constraint to produce a unique prediction across all considered horizons, reflecting the design of our method. We verify the impact of this assumption by measuring performance gains afforded by learning distinct predictions per horizon. As shown in Table 3, with full precision-recall curves in Figure 19, we find no statistical evidence for performance gain over using $\alpha^{step}$ on all tasks and studied metrics. Thus, models do not appear to leverage this additional flexibility offered by MHP. With superior results on all timestep- and event-based experiments, and greater scalability thanks to the single prediction horizon modeled, we find temporal label smoothing to be a superior training objective to MHP in early prediction tasks.

Table 3: **Do MHP's multiple outputs improve performance over TLS with $q^{step}$?** We provide $p$-values for the paired Student-t test [38] on the null hypothesis $H_0 : \mu_{step} \geq \mu_{MHP}$. With no statistically significant improvements ($p < 0.05$), we justify our assumption in Proposition 1.

| Task | Circulatory Failure | | Decompensation | | Respiratory Failure | |
|---|---|---|---|---|---|---|
| Method | AUPRC | Recall | AUPRC | Recall | AUPRC | Recall |
| MHP | $39.6 \pm 0.5$ | $30.3 \pm 1.0$ | $34.9 \pm 0.3$ | $28.6 \pm 0.5$ | $60.3 \pm 0.1$ | $76.6 \pm 0.5$ |
| TLS ($\alpha^{step}$) | $39.3 \pm 0.2$ | $29.4 \pm 0.8$ | $35.2 \pm 0.3$ | $29.2 \pm 0.4$ | $60.5 \pm 0.1$ | $77.4 \pm 0.5$ |
| p-value ($H_0$) | 0.11 | 0.10 | 0.95 | 0.97 | 0.99 | 0.98 |

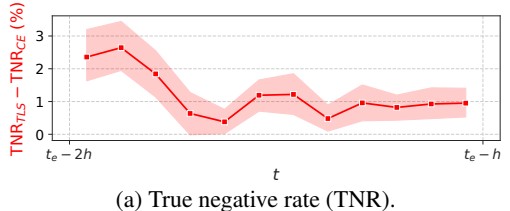 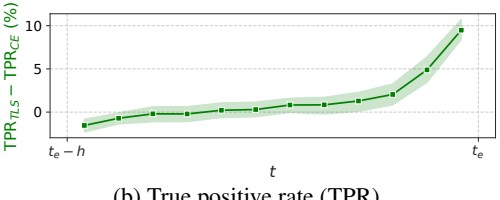

(a) True negative rate (TNR).            (b) True positive rate (TPR).

Figure 7: **Performance improvement over time** for TLS over traditional cross-entropy on circulatory failure prediction. Timestep-level metrics computed for a precision of $0.5$ over two-hour bins.

### 5.3 TRADE-OFFS AND LIMITATIONS

Despite the demonstrated advantage of our training paradigm for two distinct early prediction tasks, we observed more limited performance gain over traditional cross-entropy when predicting respiratory failure in Table 2, as with other baselines. This observation motivated an analysis of the specific problem settings in which our objective helps. Respiratory failure events are much more frequent than circulatory failure or decompensation, with the majority of ICU patients undergoing approximately two such events during their stay, as quantified in Appendix B. We hypothesize that this reduced class imbalance leads to sufficient discriminative information within the label boundary region. This belief is supported by the lower performance drop from event to prediction time in Figure 4 in comparison to other tasks, and results in a more significant performance loss close to $t_e - h$ with TLS, with a 1% drop in true positive rate (TPR) in Figure 8.

However, as expected by design, our method improves recall (+1% TPR) over cross-entropy close to the event. This also leads to a non-negligible 0.4% ($p$-value $< 0.05$) improvement in event recall, visualized in Appendix D.2. Overall, this analysis reveals that whereas TLS has little impact on global metrics for tasks with limited performance reduction over time (e.g., close-to-balanced, of often limited usefulness in clinical decision support efforts [8]), it still results in clinically meaningful performance improvements along per-horizon and event-based metrics.

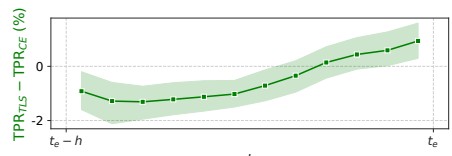

Figure 8: **Performance improvement over time** for TLS over traditional cross-entropy, for respiratory failure. True positive rates (TPR) are computed for a precision of $0.5$ over 2-hour bins.

## 6 CONCLUSION

Early prediction of adverse events is paramount to the development of clinical decision support systems, with a demonstrated potential to improve patient outcomes [3]. Still, this task remains poorly studied in the machine learning literature, with few training solutions tailored to address its challenges. Based on typically rare and noisy labels, models must learn to discriminate a predictive signal in anticipation of events to allow an adequate medical response.

After highlighting the limitations of traditional classification objectives and methods designed to address the class imbalance, we propose a novel training framework that leverages trends in event signals over time. We show that multi-horizon prediction, a heuristic used to improve early prediction, can be formalized as a restriction of our framework. Simple but effective, temporal label smoothing empirically matches or outperforms all considered baselines on various tasks and datasets, with significant improvements on clinically-relevant evaluation metrics. Performance gains are limited, as with other baselines, for respiratory failure prediction in which higher event prevalence provides sufficient informative data points for the model to learn through a conventional cross-entropy objective. In further work, we aim to explicitly adapt the temporal inductive bias to the task at hand and to combine temporal label smoothing with recent objectives designed to directly optimize AUPRC, such as minimum precision constraint [39] or dice-based loss functions [40].

Looking ahead, we expect that temporal label smoothing will be leveraged to develop more clinically reliable systems for risk prediction of rare adverse events. Further research on tailored machine learning solutions to improve real-world decision support holds promise for better clinical care and operations management.

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

# A  THEORETICAL DETAILS

## A.1  MULTI-HORIZON PREDICTION: PROOF OF PROPOSITION 1

**Equivalency between MHP and TLS objectives.**  Recalling the formalism of multi-horizon prediction outlined in Section 3.3, true labels and model predictions can be rewritten as $\mathbf{y}_{i,t} = [y_{i,t}^{h_1}, \ldots, y_{i,t}^{h}, \ldots, y_{i,t}^{h_H}]$ and $\hat{\mathbf{y}}_{i,t} = [\hat{y}_{i,t}^{h_1}, \ldots, \hat{y}_{i,t}^{h}, \ldots, \hat{y}_{i,t}^{h_H}]$, where $H$ is the number of horizons considered. The training objective for patient $i$ becomes:

$$L^{MHP}(\mathbf{y}_{i,t}, \hat{\mathbf{y}}_{i,t}) = -\frac{1}{H} \sum_{k=1}^{H} y_{i,t}^{h_k} \log(\hat{y}_{i,t}^{h_k}) + (1 - y_{i,t}^{h_k}) \log(1 - \hat{y}_{i,t}^{h_k})$$

The assumption that $\{\hat{y}_{i,t}^{h_k}\}_k$ is **equal** for all $k$ allows to rewrite the objective as follows:

$$L^{MHP}(\mathbf{y}_{i,t}, \hat{\mathbf{y}}_{i,t}) = -\left[ \log(\hat{y}_{i,t}) \frac{1}{H} \sum_{k=1}^{H} y_{i,t}^{h_k} + \log(1 - \hat{y}_{i,t}) \frac{1}{H} \sum_{k=1}^{H} (1 - y_{i,t}^{h_k}) \right]$$

with $\hat{y}_{i,t}$ being the common prediction shared across all horizons. This equation can now be viewed as a temporal label smoothing objective with smoothed labels $q^{step}(1|i,t) = \frac{1}{H} \sum_{k=1}^{H} y_{i,t}^{h_k}$:

$$L^{MHP}(\mathbf{y}_{i,t}, \hat{\mathbf{y}}_{i,t}) = -\left[ \log(\hat{y}_{i,t}) \cdot q^{step}(1|i,t) + \log(1 - \hat{y}_{i,t}) \cdot \left(1 - q^{step}(1|i,t)\right) \right]$$

**Smoothing parametrization.**  Next, we aim to recover the explicit form of $q^{step}(1|i,t)$. Without loss of generality, we assume that horizons $\{h_k\}_k$ are in ascending order. The temporal dependency between samples, formalized in Equation 1), results in the following relationship between predictions at horizons $h_u$ and $h_v$ :

$$v \leq u \quad \text{and} \quad y_{i,t}^{h_v} = 1 \implies y_{i,t}^{h_u} = 1 \tag{6}$$

$$v \geq u \quad \text{and} \quad y_{i,t}^{h_v} = 0 \implies y_{i,t}^{h_u} = 0 \tag{7}$$

Thanks to the above property, we can determine $q^{step}(1|i,t)$ by studying three cases of multi-horizon labels, illustrated in Figure 9. For notational simplicity, we define $d_e(i,t) = t_e(i,t) - t$.

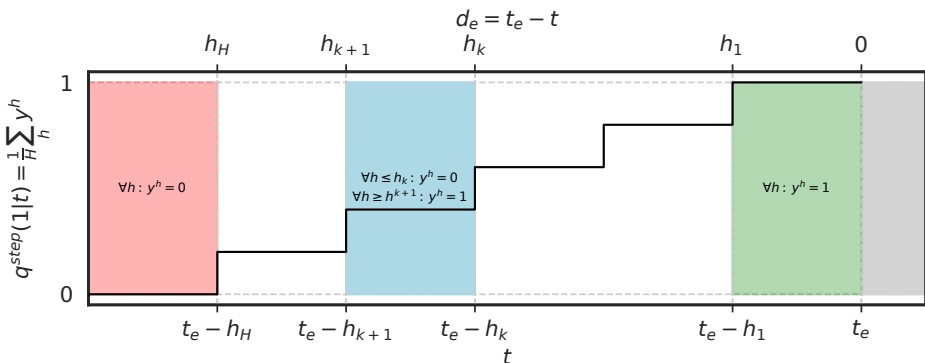

Figure 9: **Label values for multi-horizon prediction**, and conversion to smoothed labels $q^{step}(1|t)$.

**Case 1:** $d_e(i,t) \leq h_1$.
From label definition. we have that $y_{i,t}^{h_1} = 1$ if $d_e(i,t) \leq h_1$. As $h_1$ is the smallest horizon, following Equation 6, we have $y_{i,t}^{h_c} = 1, \forall c \in [\![1, H]\!]$. We can rewrite the objective as:

$$L^{MHP}(\mathbf{y}_{i,t}, \hat{\mathbf{y}}_{i,t}) = -\log(\hat{y}_{i,t})$$
$$= -[q^{step}(1|i,t) \log(\hat{y}_{i,t}) + (1 - q^{step}(1|i,t)) \log(1 - \hat{y}_{i,t})]$$

where $q^{step}(1|i,t) = 1$.

**Case 2:** $d_e(i,t) > h_H$.

Similarly, if $d_e(i,t) > h_H$, then $y_{i,t}^{h_H} = 0$ which implies $y_{i,t}^{h_c} = 0, \forall c \in [\![1,H]\!]$ from Equation 7. The objective can be rewritten as:

$$L^{MHP}(\mathbf{y}_{i,t}, \hat{\mathbf{y}}_{i,t}) = -\log(1 - \hat{y}_{i,t})$$
$$= -[q^{step}(1|i,t)\log(\hat{y}_{i,t}) + (1 - q^{step}(1|i,t))\log(1 - \hat{y}_{i,t})]$$

where $q^{step}(1|i,t) = 0$.

**Case 3:** $\exists k \in [\![1, H-1]\!]$   s.t   $h_k < d_e(t) \leq h_{k+1}$.

Following the same reasoning as in the first two cases, we now have a specific index $k$ which separates positive and negative labels. We have $y_{i,t}^{h_c} = 0, \forall c \in [\![1,k]\!]$ and $y_{i,t}^{h_c} = 1, \forall c \in [\![k+1, H]\!]$. This allows to rewrite the objective as follows:

$$L^{MHP}(\mathbf{y}_{i,t}, \hat{\mathbf{y}}_{i,t}) = -[\frac{H-k}{H}\log(\hat{y}_{i,t}) + \frac{k}{H}\log(1 - \hat{y}_{i,t})]$$
$$= -[q^{step}(1|i,t)\log(\hat{y}_{i,t}) + (1 - q^{step}(1|i,t))\log(1 - \hat{y}_{i,t})]$$

where

$$q^{step}(1|i,t) = \frac{H-k}{H}.$$

Defining a new smoothing parametrisation $\alpha^{step}$ such that $q^{step}(1|i,t) = 1 - \alpha^{step}(i,t)$, we obtain:

$$\alpha^{step}(i,t) = \begin{cases} \frac{k}{H} & \text{if} \quad h_k \leq d_e(i,t) < h_{k+1} \quad \forall k \leq H-1 \\ 0 & \text{if} \quad d_e(i,t) \leq h_1 \\ 1 & \text{if} \quad d_e(i,t) > h_H \end{cases}$$

Thus, $\forall d_e(t) > 0$, we find that $L_i^{MHP} = L_i^{TLS}$ when smoothed labels are defined as $q^{step}(1|i,t) = 1 - \alpha^{step}(i,t)$. This concludes our proof. □

## A.2   TEMPORAL LABEL SMOOTHING FUNCTIONS

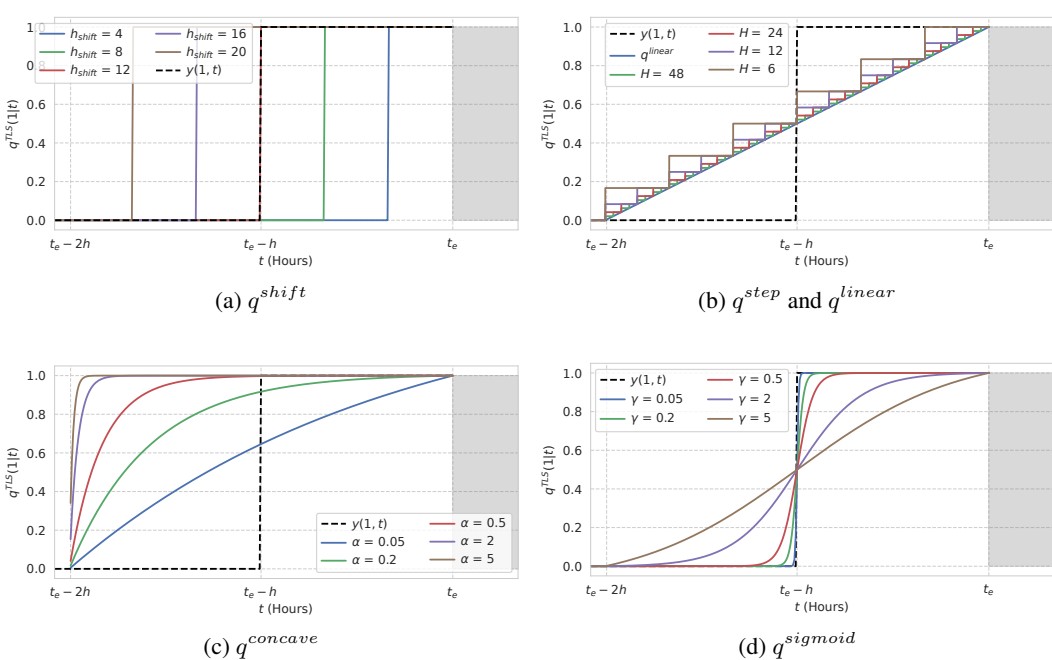

(a) $q^{shift}$

(b) $q^{step}$ and $q^{linear}$

(c) $q^{concave}$

(d) $q^{sigmoid}$

Figure 10: **Illustration of temporal label smoothing** with alternative smoothing parametrizations.

Motivated by prior work [8; 18], we compare the performance of various smoothing functions $\alpha(i, t)$. All proposed parametrizations are continuous and monotonous decreasing functions that satisfy boundary conditions $\alpha(i, t_e(i, t) - 2h) = 1$ and $\alpha(i, t_e(i, t)) = 0$. As evidenced in Table 4, we find exponential label smoothing to perform best or as well as others across all tasks and metrics. Performance as a function of hyperparameter setting can be visualized in Figure 11. All model and hyperparameter selection were carried out on the validation set, including the final choice of parametrization function.

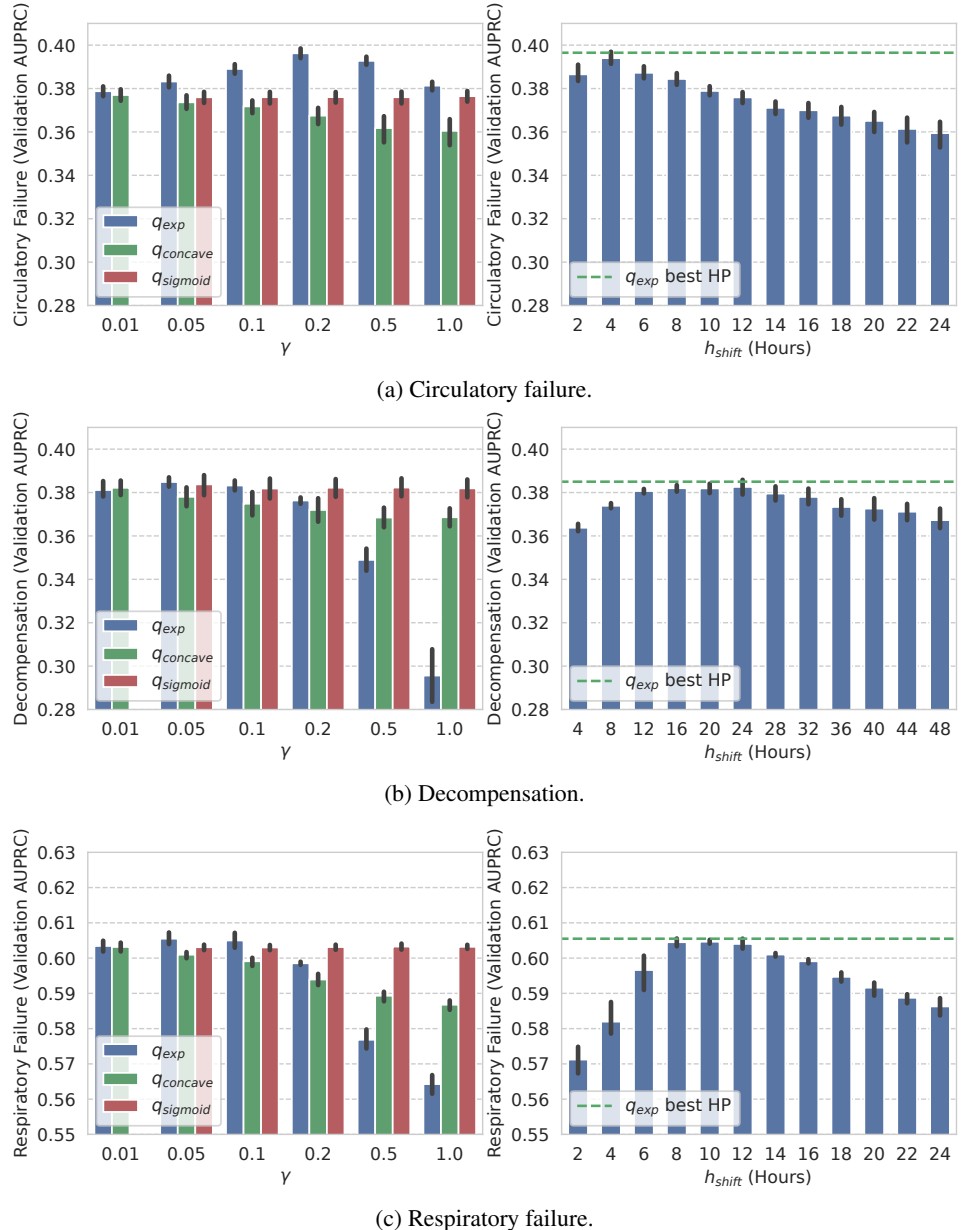

Figure 11: **Validation AUPRC performance of temporal label smoothing as a function of smoothing hyperparameters**, with different smoothing parameterizations. (Left) Performance for different smoothing strengths $\gamma$ with $\alpha^{exp}, \alpha^{concave}, \alpha^{sigmoid}$; (Right) Performance for different prediction horizons $h_{shift}$ with $\alpha^{shift}$ smoothing.

**Shifted boundary labels.** Shifting the prediction horizon or label boundary in training can be viewed as a form of temporal label smoothing, in which class labels are inverted within a prediction

Table 4: **Performance of different smoothing functions on early prediction tasks.** Recall is reported at a 50% precision.

| Task | Circulatory Failure | | Decompensation | | Respiratory Failure | |
|---|---|---|---|---|---|---|
| Method | AUPRC | Recall | AUPRC | Recall | AUPRC | Recall |
| $\alpha^{step}$ | $39.3 \pm 0.2$ | $29.4 \pm 0.8$ | $35.2 \pm 0.3$ | $29.2 \pm 0.4$ | $60.5 \pm 0.1$ | $77.4 \pm 0.5$ |
| $\alpha^{shift}$ | $40.1 \pm 0.3$ | $31.8 \pm 0.6$ | $34.5 \pm 0.4$ | $28.2 \pm 0.5$ | $60.5 \pm 0.2$ | $77.3 \pm 0.5$ |
| $\alpha^{linear}$ | $39.4 \pm 0.3$ | $29.7 \pm 0.8$ | $35.1 \pm 0.4$ | $29.2 \pm 0.6$ | $60.3 \pm 0.3$ | $77.0 \pm 0.6$ |
| $\alpha^{sigmoid}$ | $39.4 \pm 0.3$ | $29.7 \pm 0.8$ | $34.9 \pm 0.4$ | $28.8 \pm 0.5$ | $60.6 \pm 0.2$ | $77.3 \pm 0.5$ |
| $\alpha^{concave}$ | $39.4 \pm 0.3$ | $29.7 \pm 0.8$ | $35.1 \pm 0.4$ | $29.2 \pm 0.6$ | $60.3 \pm 0.3$ | $77.0 \pm 0.6$ |
| $\alpha^{exp}$ | $\mathbf{40.6 \pm 0.3}$ | $\mathbf{32.3 \pm 0.7}$ | $\mathbf{35.5 \pm 0.3}$ | $\mathbf{29.3 \pm 0.4}$ | $60.4 \pm 0.2$ | $77.0 \pm 0.3$ |

window of interest. This defines the following smoothing parametrization $\alpha^{shift}(i, t)$:

$$\alpha^{shift}(i, t) = \mathbb{1}\left[t_e(i, t) - t \geq h_{shift}\right] \tag{8}$$

where $h_{shift}$ is a hyperparameter controlling the horizon of the smoothed labels ($h_{shift} = h$ corresponds to cross-entropy training). The strength of this smoothing function is illustrated in Figure 10a.

Figure 11 outlines the performance of this alternative smoothing parametrization as a function of $h_{shift}$. For both decompensation and respiratory failure, shifting the label boundary closer to the event time decreases performance. On circulatory failure, performance does improve over traditional cross-entropy training as the label horizon is brought closer to the event of interest, which can be interpreted as an inductive bias similar to that induced by the exponential smoothing function.

**Linear label smoothing.** The most straightforward extension to the step function $\alpha^{step}$ described in Section 3.3 is a linear label smoothing corresponding to the case $H \to +\infty$.
Our parametrization $\alpha^{linear}(i, t)$ is thus defined as follows:

$$\alpha^{linear}(i, t) = \begin{cases} \frac{t_e(i,t) - t}{2h} & \text{if} \quad t_e(i, t) - t < 2h \\ 1 & \text{if} \quad t_e(i, t) - t \geq 2h \end{cases} \tag{9}$$

We illustrate the impact of the number of steps $H$ in Figure 10b.

**Sigmoidal label smoothing.** Another natural direction to explore is to smooth labels starting from the true distribution, a unique step function at $t = t_e(t) - h$. This can be achieved by defining $\alpha(t)$ as a generalized logistic function [41]:

$$\alpha^{sigmoid}(i, t) = \begin{cases} 1 - \frac{K - A}{1 + e^{\frac{t_e(i,t) - t - d}{\gamma}}} - A & \text{if } t_e(i, t) - t < 2h \\ 1 & \text{if } t_e(i, t) - t \geq 2h \end{cases} \tag{10}$$

where $K$, $A$ and $d$ are three constants fixed by imposing the boundary conditions at $t = t_e(i, t) - 2h$ and $t = t_e(i, t)$, as well as $\alpha(t_e(i, t) - 2h) = \frac{1}{2}$. This yields:

$$K = -A e^{\frac{2h - d}{\gamma}}$$

$$A = \frac{e^{\frac{-d}{\gamma}} + 1}{e^{\frac{-d}{\gamma}} - e^{\frac{2h - d}{\gamma}}}$$

$$d = h$$

As shown in Figure 10d, $\gamma$ controls the smoothing strength, interpolating between the true distribution $\delta_{y_i=1}$ as $\gamma \to 0$ and $q^{linear}$ when $\gamma \to +\infty$.

**Exponential label smoothing.**   The smoothing function we find to perform best is the exponential decay one. This idea is motivated by survival analysis, where patient survival probability can be modeled as the exponential decay of a cumulative hazard function [18; 42]. In practice, as defined in Section 3.2, our exponential smoothing function $\alpha^{exp}(i,t)$ is defined as follows:

$$\alpha^{exp}(i,t) = \begin{cases} 1 - e^{-\gamma(t_e(i,t)-t-d)} - A & \text{if } t_e(i,t) - t < 2h \\ 1 & \text{if } t_e(i,t) - t \geq 2h \end{cases} \quad (11)$$

where parameters $\{d, A\}$ are set to satisfy boundary conditions:

$$A = -e^{-\gamma(2h-d)}$$

$$d = -\frac{1}{\gamma} \ln\left(1 - e^{-\gamma 2h}\right)$$

Here, $\gamma$ also controls the smoothing strength between $q^{linear}$ when $\gamma \to 0$ and $q(t) = 0 \,\forall t < t_e$ when $\gamma \to +\infty$.

Overall, despite $\alpha^{sigmoid}$ and $\alpha^{shift}$ achieving good results on respiratory and circulatory failure respectively, $\alpha^{exp}$ statistically outperforms these smoothing parameterizations across all tasks on validation metrics. An interesting avenue for further work would be to combine exponential smoothing with the boundary shift approach, or effectively change $(h_{min}, h_{max})$, which was fixed to $(0, 2h)$ in our work for a fair comparison to multi-horizon prediction.

**Concave exponential label smoothing.**   Finally, to mirror the behavior of the exponential smoothing function away from linear interpolation and investigate its effect on performance, we designed the following concave smoothing function $\alpha^{concave}$:

$$\alpha^{concave}(i,t) = \begin{cases} e^{-\gamma(d-t_e(i,t)+t)} - A & \text{if } t_e(i,t) - t < 2h \\ 1 & \text{if } t_e(i,t) - t \geq 2h \end{cases} \quad (12)$$

Parameters $\{d, A\}$ are identical to the convex smoothing function parameters, set to satisfy boundary conditions. The strength of this concave smoothing function is illustrated Figure 10c.

No performance gains were obtained through temporal label smoothing with a concave function, as shown in Figure 11. This smoothing function effectively penalizes false positives harder than false negatives, which is less adapted to our tasks of interest (in contrast to the convex $a^{exp}$). As a result, the best-performing concave parametrization is consistently obtained with the lowest value of $\gamma$, closer to a linear function choice.

## A.3   RELATED TIME-SERIES TASKS

**Comparison to survival analysis.**   Survival analysis consists of statistical methods concerned with predicting the probability of a certain event taking place over time [42]. In our formalism outlined in Section 3.1, the corresponding task is to regress the time of the next event, $t_e(t,i)$, based on patient information accumulated up to time $t$. To recover early event prediction, a threshold on the hazard model can thus be applied to determine whether an event will happen within our horizon of interest $h$. Modeling constraints imposed in survival analysis improve time-to-event prediction performance over traditional regression methods, which supports our approach to leverage the temporal structure of our comparable task. Interestingly, recent developments in survival modeling to deal with dynamic predictions have been addressed with multi-horizon prediction [43].

Still, distinctions must be highlighted between our adverse event prediction problem and the typical experimental setup for survival analysis: in our case, multiple events can occur over the course of a patient's stay, with unknown patient states during and immediately after event occurrence. This results in complex, informative censoring patterns and challenges common assumptions in survival analysis, which can therefore not be directly applied to our task.

**Comparison to early time-series classification.**   A distinction must be drawn between our task of early prediction of adverse events and that of early time-series classification. The latter has been more extensively explored in the literature [44; 45; 46], but addresses a distinct problem.

Considering a time series up to timestep $t$, early event prediction is concerned with classifying *whether* a particular event will occur between $t$ and $t + h$, for a fixed horizon $h$. Predictions are made at each timepoint over the entire time series: as multiple samples arise from the same time series and therefore depend on one another over time, these should not be considered as i.i.d.

In contrast, early classification of time series aims to regress *the first timepoint $t$* at which a label for the entire time series can be predicted with a desired accuracy [44]. A single prediction is made, as soon as possible, for the entire series – which can be considered an independent sample from the dataset of time series. This latter task can be framed as early prediction of the event "prediction is possible", where $h = \infty$, given a separate time-series classifier. As a result, an interesting avenue of further work would be to apply temporal label smoothing to the latter task.

On the other hand, early event prediction cannot be translated into a simple early classification problem. As a result, methods designed for early time-series classification are therefore not applicable to this problem setting.

### A.4    BASELINE OBJECTIVE FUNCTIONS

In this section, we clarify the mathematical formalism behind our baselines to facilitate comparison to temporal label smoothing. All baselines explored effectively propose a modification of the cross-entropy objective often used for binary classification tasks, $L_i = L^{CE}(y_i, \hat{y}_i)$.

**Balanced cross-entropy.**    To facilitate learning from highly imbalanced datasets, balanced cross-entropy relies on reweighting samples based on their class prevalence, as follows:

$$L^{CE} = \frac{1}{N} \sum_i^N \omega_{y_i} L(\hat{y}_i, y_i) \tag{13}$$

where $C$ is the number of classes, $\omega_{y_i} = \frac{1}{C \cdot b(y_i)}$ and $b(c)$ defines the prevalence of class $c$ such that $\sum_c b(c) = 1$. Regular cross-entropy corresponds to the case where $b(c) = \frac{1}{C}$ for all classes. In the binary setting, $b(1)$ can be treated as a hyperparameter determining the contribution of the minority class to the loss.

**Focal loss.**    Denoting our output prediction as $\hat{y}_i = p_\theta(y_i = 1)$, the focal loss objective for binary classification of target $y_i$ is a variant on the balanced cross-entropy loss:

$$L^{focal}(y_i, \hat{y}_i) = -\omega_1 (1 - \hat{y}_i)^\zeta y_i \log(\hat{y}_i) - \omega_0 \hat{y}_i^\zeta (1 - y_i) \log(1 - \hat{y}_i)$$

where $\omega_{y_i}$ is a balancing weight for class $y_i$ and $\zeta$ is the focal loss weight.

**Multi-horizon prediction.**    As highlighted in Section 3.3, multi-horizon training can be formalized as the following objective:

$$L^{MHP}(\mathbf{y}_{i,t}, \hat{\mathbf{y}}_{i,t}) = -\frac{1}{H} \sum_{k=1}^H y_{i,t}^{h_k} \log(\hat{y}_{i,t}^{h_k}) + (1 - y_{i,t}^{h_k}) \log(1 - \hat{y}_{i,t}^{h_k})$$

where true labels and model predictions are given by $\mathbf{y}_{i,t} = [y_{i,t}^{h_1}, \ldots, y_{i,t}^h, \ldots, y_{i,t}^{h_H}]$ and $\hat{\mathbf{y}}_{i,t} = [\hat{y}_{i,t}^{h_1}, \ldots, \hat{y}_{i,t}^h, \ldots, \hat{y}_{i,t}^{h_H}]$, for $H$ distinct horizons.

**Label smoothing.**    As introduced by Szegedy et al. [14], label smoothing consists of substituting the original label distribution $\delta_{y_i = c}$ in the cross-entropy objective $L_i = L^{CE}(y_i, \hat{y}_i)$ by a smoothed version $q^{LS}(c|y_i)$. This surrogate distribution over classes $c$ is defined as follows :

$$q^{LS}(c|y_i) = \delta_{y_i = c}(1 - \alpha) + u(c)\alpha \tag{14}$$

In the original approach, $u$ is uniform and $\alpha \in [0, 1]$ controls the smoothing strength. By shifting the minimum of the objective function away from $\hat{y}_i = 1$, labels smoothing prevents the model from becoming overconfident during training. Alternative designs for $u$ have been proposed [28; 29; 30] but are incompatible with the binary nature of adverse event prediction. In binary tasks, labeling is

defined according to the positive class such that $y_i \in \{0, 1\}$ and $\hat{y}_i = p_\theta(y_i = 1)$. Label smoothing therefore becomes a linear interpolation with parameter $\alpha$ such that $q^{LS}(1|y_i) = p(y_i = 1)$:

$$q^{LS}(1|y_i) = (1 - \alpha)y_i + \alpha(1 - y_i) \qquad (15)$$

As suggested by Lukasik et al. [27], label smoothing can be used to regularize early prediction models due to the inherently noisy nature of the task. It does not, however, account for the time dependency between samples of a given stay – highlighted in our problem formalism (Section 3.1). In contrast, temporal label smoothing modulates smoothing based on time $t$ to infuse this prior knowledge into the training objective.

## B  DATASET DETAILS

### B.1  TASK DEFINITION

In this section, we provide more details on the definition of our early prediction tasks for circulatory failure and respiratory failure from HiB [20] and decompensation from M3B [32]. A breakdown of event prevalence for each clinical endpoint is given in Table 5.

Table 5: **Event prevalence analysis**, highlighting class imbalance. Positive timesteps are counted for 12-hour and 24-hour horizons for HiRID tasks and decompensation respectively. Statistics are computed on the training set.

| Task | Positive timesteps (%) | Patients undergoing event (%) | Number of events per positive patient |
|------|------------------------|-------------------------------|---------------------------------------|
| Circulatory Failure (HiRID) | 4.3 | 25.6 | 1.9 |
| Respiratory Failure (HiRID) | 38.6 | 83.0 | 1.8 |
| Decompensation (MIMIC) | 2.1 | 8.3 | 1.0 |

**Circulatory failure** is a failure of the cardiovascular system, detected in practice through elevated arterial lactate ($> 2$ mmol/l) and either low mean arterial pressure ($< 65$ mmHg) or administration of a vasopressor drug. Yèche et al. [20] defines a patient to be experiencing a circulatory failure event at a given time if those conditions are met for $2/3$ of time points in a surrounding two-hour window. Early prediction labels are then derived from these event labels as outlined in Section 3.1.

**Respiratory failure** is defined by Yèche et al. [20] as a P/F ratio (arterial $pO_2$ over $FIO_2$) below 300 mmHg. This definition includes mild respiratory failure, which explains higher event prevalence in Table 5. As above, Yèche et al. [20] consider a patient to be experiencing respiratory failure if $2/3$ of timepoints are positive within a surrounding 2h window.

**Decompensation** refers to the death of a patient. Event labels are directly extracted from the MIMIC-III [33] metadata about the time of death of a patient. Early prediction labels are also extracted following Section 3.1. Note that decompensation can occur outside of the ICU stay if a patient is sent to a palliative unit, for instance, which can result in patient stays with fewer than 24 positive samples.

### B.2  PRE-PROCESSING

We describe the pre-processing steps we applied to both datasets, HiRID and MIMIC-III.

**Imputation.**  Diverse imputation methods exist for ICU time series. For simplicity, we follow the approach of original benchmarks [32; 20] by using forward imputation when a previous measure existed. The remaining missing values are zero-imputed after scaling, corresponding to a mean imputation.

**Scaling.**  Whereas prior work explored clipping the data to remove potential outliers [8], we do not adopt this approach as we found it to reduce performance on early prediction tasks. A possible explanation is that, due to the rareness of events, clipping extreme quantiles may remove parts of the signal rather than noise. Instead, we simply standard-scale data based on the training sets statistics.

## C IMPLEMENTATION DETAILS

**Training details.** For all models, we set the batch size according to the available hardware capacity. Because transformers are memory-consuming, we train the models for respiratory failure and decompensation with a batch size of 8 stays. On the other hand, we train the GRU model for circulatory failure with a batch size of 64. We early stopped each model training according to their validation loss when no improvement was made after 10 epochs.

**Libraries.** A full list of libraries and the version we used is provided in the `environment.yml` file. The main libraries on which we build our experiments are the following: pytorch 1.11.0 [47], scikit-learn 0.24.1[48], ignite 0.4.4, CUDA 10.2.89[49], cudNN 7.6.5[50], gin-config 0.5.0 [51].

**Infrastructure.** We follow all guidelines provided by `pytorch` documentation to ensure the reproducibility of our results. However, reproducibility across devices is not ensured. Thus we provide here the characteristics of our infrastructure. We trained all models on a single `NVIDIA RTX2080Ti` with a `Xeon E5-2630v4` core. Training took between 3 and 10 hours for a single run.

**Uncertainty estimation.** We compute uncertainty estimates over a population of 10 training instances with different seeds. This widely-used approach has the advantage to account for the stochasticity of the training procedure, which we found to be predominant in early prediction tasks. This approach differs from other work [25; 23; 8; 24] which computes uncertainty estimate by bootstrapping the test population. We compare both approaches in Appendix D.4 to demonstrate that using a pivot bootstrap estimator decreases confidence intervals by effectively increasing the population size. To be conservative with our results, we retained the former approach to compute statistics across 10 training instances. We report the 95% confidence interval over the population means in all experiments.

**Architecture choices** We used the same architecture and hyperparameters reported giving the best performance on respiratory and circulatory failure in Yèche et al. [20]. For these tasks, we only optimized embedding regularization parameters [8]. Exact parameters are reported in Table 6 and Table 7. For decompensation, as we found a transformer architecture to perform better than originally proposed models [32], we carried out our own random search on validation AUPRC performance. Exact parameters for this task are reported in Table 8.

Table 6: **Hyperparameter search range** for circulatory failure with GRU [34] backbone. In **bold** are parameters selected by random search.

| Hyperparameter | Values |
|---|---|
| Learning Rate | (1e-5, 3e-5, 1e-4, **3e-4**) |
| Drop-out | (0.0, 0.1, **0.2**, 0.3, 0.4) |
| Depth | (1, **2**, 3) |
| Hidden Dimension | (32, 64, 128, **256**) |
| L1 Regularization | (1e-2, 1e-1, 1, **10**) |

### C.1 BASELINE IMPLEMENTATION

**Balanced cross-entropy.** In the binary setting, the only hyperparameter of balanced cross-entropy is the relative contribution of the minority class to the loss, $\omega_1$. As discussed in Section 5.2, no value of $\omega_1$ was found to improve validation performance over the non-balanced case $\omega_1 = 1$.

**Focal loss.** A grid search over focal loss hyperparameters was also carried out. Similarly to balanced cross-entropy, on all tasks, no values of focal loss weight $\zeta$ or balancing weight $\omega_1$ were found to outperform regular cross-entropy corresponding to $\zeta = 0$ and $\omega_1 = 1$.

Table 7: **Hyperparameter search range** for respiratory failure with Transformer [35] backbone. In **bold** are parameters selected by random search.

| Hyperparameter | Values |
|---|---|
| Learning Rate | (1e-5, 3e-5, **1e-4**, 3e-4) |
| Drop-out | (0.0, 0.1, 0.2, **0.3**, 0.4) |
| Attention Drop-out | (**0.0**, 0.1, 0.2, 0.3, 0.4) |
| Depth | (1, **2**, 3) |
| Heads | (**1**, 2, 4) |
| Hidden Dimension | (32, **64**, 128, 256) |
| L1 Regularization | (1e-2, 1e-1, 1, **10**) |

Table 8: **Hyperparameter search range** for decompensation with Transformer [35] backbone. In **bold** are parameters selected by random search.

| Hyperparameter | Values |
|---|---|
| Learning Rate | (1e-5, 3e-5, **1e-4**, 3e-4) |
| Drop-out | (0.0, 0.1, 0.2, **0.3**, 0.4) |
| Attention Drop-out | (0.0, **0.1**, 0.2, 0.3, 0.4) |
| Depth | (1, **2**, 3) |
| Heads | (**1**, 2, 4) |
| Hidden Dimension | (32, **64**, 128, 256) |
| L1 Regularization | (1e-2, **1e-1**, 1, 10) |

**Multi-horizon prediction.** Following Tomašev et al. [8], we consider $H$ horizons on both side of the true horizon $h$ between $0$ and $2h$. As we didn't find $H \longrightarrow +\infty$, to increase performance, we selected $H = 11$ (including true horizon $h$) compared to $H = 8$ in Tomašev et al. [8], which we found to perform slightly worse. This means we made a prediction every 2 hours for HiB tasks and every 4 hours for decompensation.

**Label Smoothing.** Label smoothing [14], as defined in Section 3.2, is normally used in multi-class setting. We still compared our method to it for two reasons. First, to explore if it can help when dealing with a noisy signal as we claim is the case for early event detection. Second, to ablate the impact of adding a temporal dependency to the method. Again, we select the hyperparameter $\alpha$ through a grid search. Interestingly, we found label smoothing to slightly improve performance over the validation set for all tasks as opposed to the results reported for the test set in Table 2. We found $\alpha = 0.05$ to perform best for circulatory failure and decompensation. For respiratory failure, we found $\alpha = 0.1$ to have the best validation performance.

## C.2  TLS IMPLEMENTATION

TLS depends on two components, the temporal range over which we smooth labels, defined by $h_{min}$ and $h_{max}$, and the smoothing function $\alpha(i, t)$. Concerning the temporal range, for a fair comparison, we fix it to match MHP, thus for all experiments we set $h_{min} = 0$ and $h_{max} = 2h$. For the smoothing function, we perform a grid search over the type of function discussed in Appendix A.2 and the smoothing strength parameter $\gamma$. For all experiments, we found $\alpha^{exp}$ to outperform other considered functions. Given validation performance, we used $\gamma = 0.2$ for circulatory failure and $\gamma = 0.05$ for respiratory failure and decompensation.

```python
def get_smoothed_labels(event_label_patient, smoothing_fn, h_true, h_min,
                        h_max, **kwargs):

    # Find when event label changes
    diffs = np.concatenate([np.zeros(1),
            event_label_patient[1:] - event_label_patient[:-1]], axis=-1)
    pos_event_change = np.where((diffs == 1) & (event_label_patient ==
        1))[0]

    # Handle patients with no events
    if len(pos_event_change) == 0:
        pos_event_change = np.array([np.inf])

    # Compute distance to closest event for each time point
    time_array = np.arange(len(event_label_patient))
    dist_all_event = pos_event_change.reshape(-1, 1) - time_array
    dist_to_closest = np.where(dist_all_event > 0,
                        dist_all_event, np.inf).min(axis=0)

    return smoothing_fn(dist_to_closest, h_true=h_true, h_min=h_min,
        h_max=h_max,
                                              **kwargs)
```

Figure 12: **Temporal label smoothing algorithm.** Python-style code to obtain smooth early prediction labels from event labels.

As discussed in Section 3.2, contrary to MHP, TLS does not require any change to the architecture leading to a computational overhead. The smoothing of the labels can be easily integrated into the data loader, as shown in Figure 12.

## D    ADDITIONAL EXPERIMENTS AND ABLATION STUDIES

This section provides additional results and experiments to complete our findings from the main manuscript. Unless otherwise stated, mean results are shown with a 95% confidence interval on the mean shaded or in error bars.

### D.1    EVENT-BASED METRICS FOR OTHER TASKS

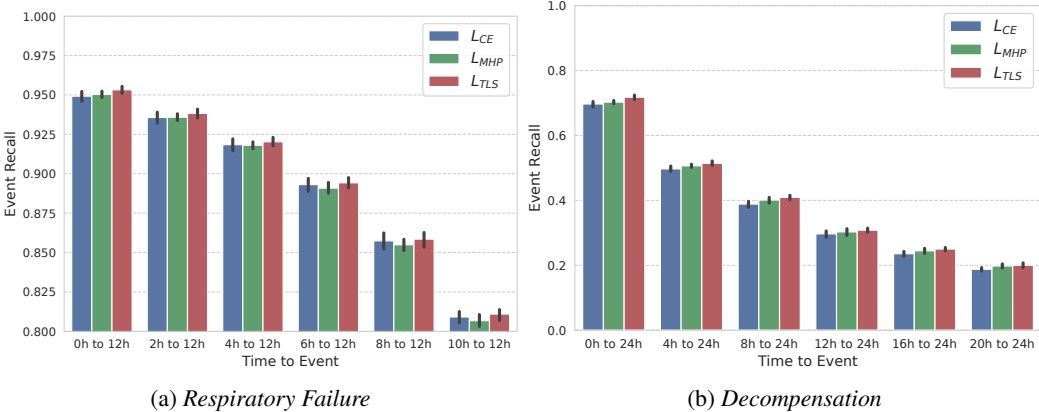

(a) *Respiratory Failure*                    (b) *Decompensation*

Figure 13: **Event recall** at 50% timestep-level precision, for two additional tasks.

Event-level performance trends for decompensation and respiratory failure prediction were similar to those obtained for circulatory failure in Figure 6b. As discussed in Section 5.1, temporal label

smoothing improves recall of adverse event episodes over cross-entropy and MHP. If the improvement observed over the baselines in terms of event-recall between 0 and $h$ are smaller than for circulatory failure, in both tasks TLS improvements over both baselines are statistically significant with paired Student's $p-$values below 0.05 for the hypothesis $\mu_{TLS} > \mu_{baseline}$.

## D.2 TIMESTEP-BASED METRICS FOR OTHER TASKS

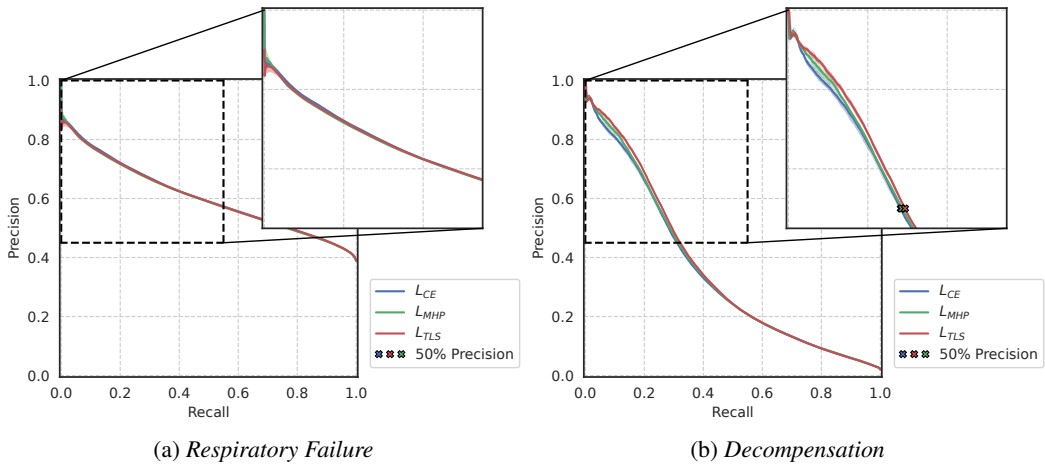

Figure 14: **Precision-recall curves**, for two additional tasks. Inset shows the clinically-applicable region with precision greater than 0.5.

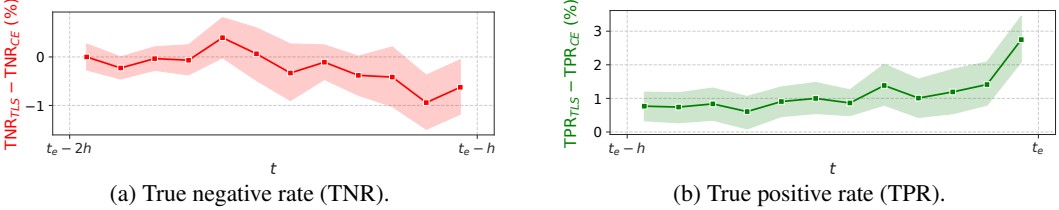

Figure 15: **Performance improvement over time** for TLS over traditional cross-entropy on decompensation prediction. Timestep-level metrics computed for precision of 0.5 over two-hour bins.

**Decompensation.** Precision-recall curves obtained for timestep-level event prediction on respiratory failure and decompensation tasks are given in Figure 14. As for circulatory failure prediction, decompensation recall gains are concentrated in regions of low false-alarm rates (>50% precision) which are most clinically relevant. Likewise, whereas recall near the label boundary $t_e - h$ is slightly negatively affected by temporal label smoothing in Figure 15, true positive rates are significantly improved leading up to the event time $t_e$. This mirrors the temporal smoothing pattern which favors higher model confidence away from the label boundary. As discussed in Section 5.2, this is aligned with clinical priorities in terms of model performance, as it ensures imminent events are better predicted.

**Respiratory Failure.** As discussed in Section 5.3, on respiratory failure, there is no clear advantage of using temporal label smoothing (or any baseline) over cross-entropy on timestep level metrics as in Figure 14. This can be attributed to the more balanced nature of this task. Still, we find that performance over time in Figure 16 reflects the design of temporal label smoothing, as true positive rates are negatively affected near the highly smoothed label boundary but improve when approaching event time.

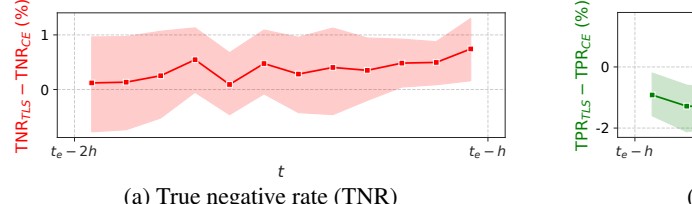
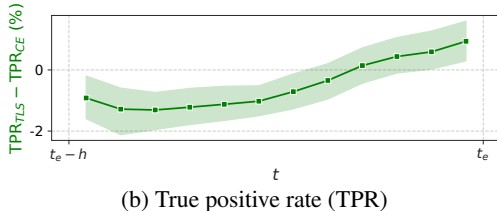

| (a) True negative rate (TNR) | (b) True positive rate (TPR) |

Figure 16: **Performance improvement over time** for TLS over traditional cross-entropy on respiratory failure prediction. Timestep-level metrics computed for precision of $0.5$ over two-hour bins.

### D.3 SUB-GROUP ANALYSIS

Populations in the intensive care unit are often heterogeneous. This has motivated recent works to focus on the fairness of deep learning across these sub-populations. In this analysis, we ensure that temporal label smoothing does not negatively affect performance in specific subgroups, compared to the objectives commonly used in the literature [8; 7; 11]. To achieve this, we measured event prediction performance across genders and age groups (below 50, between 50 and 70, and over 70 years old). As shown in Table 9, TLS matches or outperforms baseline performance across all studied subgroups, suggesting that the overall population-wide improvements are not achieved by disproportionally favouring specific cohorts. While some algorithmic bias can be observed across all methods, for instance in poorer decompensation performance amongst female patients, TLS does not appear to be amplifying this issue. In further work, we look forward to extending this analysis to more specific subgroups and to study the fairness of early event prediction methods for clinical applications.

Table 9: **Sub-group performance analysis.** We color in green improvement above the 95% confidence interval and in orange differences within it, often within the confidence interval of cross-entropy (CE).

| Circulatory Failure | Age $\leq$ 50 | | $50 <$ Age $\leq 70$ | | Age $> 70$ | | Female | | Male | |
|---|---|---|---|---|---|---|---|---|---|---|
| Method | AUPRC | Recall | AUPRC | Recall | AUPRC | Recall | AUPRC | Recall | AUPRC | Recall |
| CE | $40.4 \pm 0.5$ | $29.4 \pm 0.6$ | $38.8 \pm 0.6$ | $29.6 \pm 1.1$ | $39.2 \pm 0.3$ | $29.0 \pm 1.0$ | $39.3 \pm 0.6$ | $30.0 \pm 0.7$ | $39.1 \pm 0.4$ | $29.0 \pm 1.0$ |
| TLS | $40.4 \pm 0.5$ | $32.7 \pm 1.0$ | $41.1 \pm 0.4$ | $32.6 \pm 0.7$ | $40.0 \pm 0.3$ | $31.7 \pm 0.7$ | $41.2 \pm 0.3$ | $32.8 \pm 0.6$ | $40.4 \pm 0.3$ | $32.0 \pm 0.8$ |
| $\Delta$(TLS-CE) | 0.0 | + 3.3 | + 2.3 | + 3.0 | + 0.9 | + 2.7 | + 1.8 | + 2.9 | + 1.3 | + 3.0 |

| Respiratory Failure | Age $\leq$ 50 | | $50 <$ Age $\leq 70$ | | Age $> 70$ | | Female | | Male | |
|---|---|---|---|---|---|---|---|---|---|---|
| Method | AUPRC | Recall | AUPRC | Recall | AUPRC | Recall | AUPRC | Recall | AUPRC | Recall |
| CE | $52.1 \pm 0.2$ | $70.3 \pm 0.8$ | $62.5 \pm 0.3$ | $78.0 \pm 0.5$ | $63.2 \pm 0.3$ | $81.7 \pm 0.8$ | $54.2 \pm 0.2$ | $72.9 \pm 0.8$ | $63.7 \pm 0.2$ | $79.8 \pm 0.6$ |
| TLS | $51.9 \pm 0.4$ | $70.5 \pm 0.7$ | $62.4 \pm 0.2$ | $77.8 \pm 0.3$ | $63.2 \pm 0.3$ | $81.1 \pm 0.6$ | $53.9 \pm 0.2$ | $72.3 \pm 0.6$ | $63.7 \pm 0.2$ | $79.8 \pm 0.4$ |
| $\Delta$(TLS-CE) | - 0.3 | + 0.3 | - 0.1 | - 0.3 | 0.0 | - 0.6 | - 0.3 | - 0.6 | - 0.1 | 0.0 |

| Decompensation | Age $\leq$ 50 | | $50 <$ Age $\leq 70$ | | Age $> 70$ | | Female | | Male | |
|---|---|---|---|---|---|---|---|---|---|---|
| Method | AUPRC | Recall | AUPRC | Recall | AUPRC | Recall | AUPRC | Recall | AUPRC | Recall |
| CE | $29.2 \pm 0.8$ | $25.3 \pm 1.2$ | $34.9 \pm 0.9$ | $27.4 \pm 0.6$ | $35.8 \pm 0.2$ | $29.4 \pm 0.6$ | $30.9 \pm 0.4$ | $24.8 \pm 0.6$ | $38.3 \pm 0.6$ | $31.4 \pm 0.5$ |
| TLS | $30.5 \pm 0.5$ | $26.2 \pm 1.1$ | $36.7 \pm 0.5$ | $29.1 \pm 0.5$ | $36.3 \pm 0.3$ | $30.3 \pm 0.4$ | $31.6 \pm 0.3$ | $25.7 \pm 0.5$ | $39.6 \pm 0.5$ | $32.8 \pm 0.6$ |
| $\Delta$(TLS-CE) | + 1.3 | + 1.0 | + 1.8 | + 1.7 | + 0.5 | + 0.9 | + 0.7 | + 0.9 | + 1.3 | + 1.4 |

### D.4 UNCERTAINTY ESTIMATION WITH PIVOT BOOTSTRAP

As mentioned in Appendix C, our uncertainty estimation approach was based on measuring standard error across 10 training runs. With the uncertainty evaluation framework from Tomašev et al. [8], bootstrapping patients from our test set 200 times for each training instance, we obtained similar means to Table 2 but with confidence intervals all smaller than or equal to 0.1%. Variance within bootstrap samples from the same training instance is therefore much smaller than across instances. Our alternative uncertainty estimation approach, measuring variability between training runs, returns more conservative estimates, and was thus chosen for all results reported in this work.

### D.5 LOSS REWEIGHTING METHODS

Hyperparameter grid search results for different loss reweighting methods are shown in Figures 5 and 17. For all three tasks, both weighted cross-entropy and focal loss were found to negatively affect

performance in comparison to traditional cross-entropy. Likely explanations for these results are provided in Section 5.2: focal loss focuses training on noisily labeled samples, and weighted cross-entropy largely reduces precision. We validate the latter hypothesis by visualizing precision-recall curves of models trained with this objective in Figure 18.

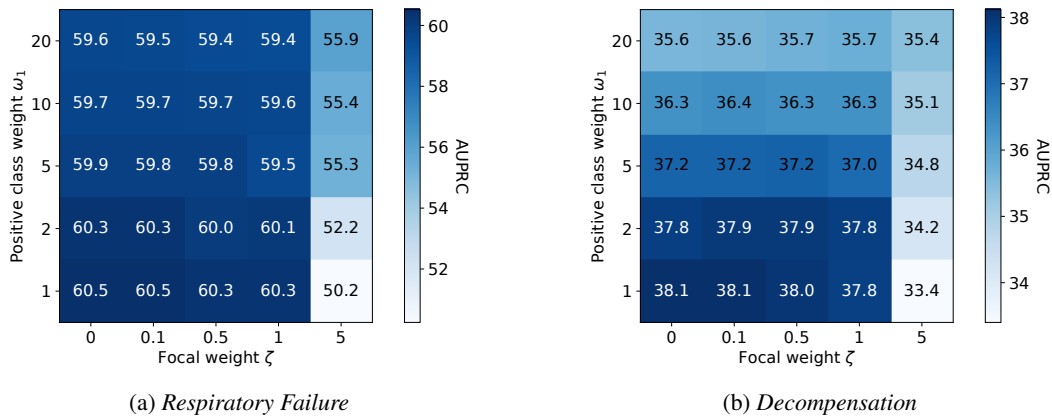

(a) *Respiratory Failure*                    (b) *Decompensation*

Figure 17: **Performance loss with class reweighting methods**, on validation set. Balanced cross-entropy corresponds to $\zeta = 0$.

**Impact of weighted cross-entropy on precision.** With a relative weight for the positive class $\omega_1 = \frac{0.5}{b(c=1)} > 1$, weighted cross-entropy encourages a greater number of true positives to improve recall. Doing so also increases the of false positives, impairing precision. In Figure 18, as the starting precision of all cross-entropy models is poor, no discernible improvements in the recall can be observed as class weights are increased, whereas precision is markedly reduced in low-recall regions. This explains the overall reduction in AUPRC with this method across all tasks.

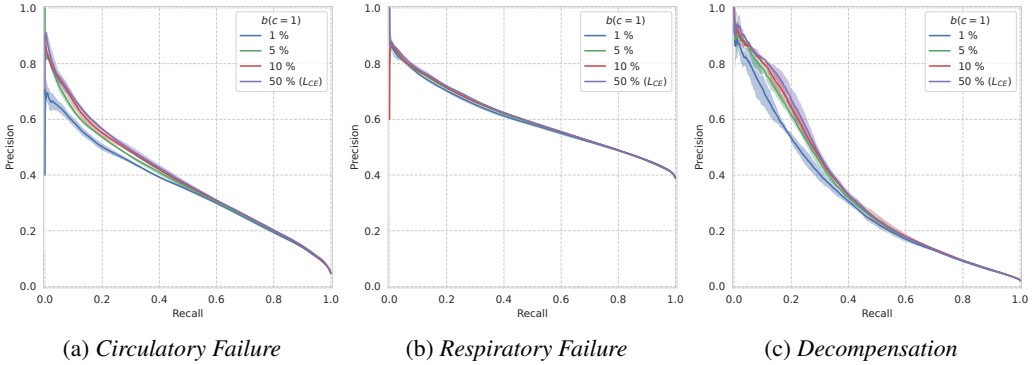

(a) *Circulatory Failure*          (b) *Respiratory Failure*          (c) *Decompensation*

Figure 18: **Class reweighting impact on AUPRC**. Class reweighting does not improve AUPRC because it significantly reduces precision. Balance weights correspond to $b(c)$.

### D.6    VISUAL COMPARISON OF TLS WITH $q_{step}$ AND MHP PERFORMANCE

In Figure 19, we compare the precision-recall curve of multi-horizon prediction and temporal label smoothing with $q_{step}$ smoothing, ensuring that there is no area where MHP is superior. In complement to Table 3 and to the analysis in Section 5.2, this confirms that predicting a single horizon with a step function smoothing is sufficient to match the performance of multi-horizon prediction.

### D.7    COMBINING TLS WITH OTHER METHODS

Finally, we investigated whether temporal label smoothing could be combined with other objective functions to leverage their respective added value and further improve prediction performance. The

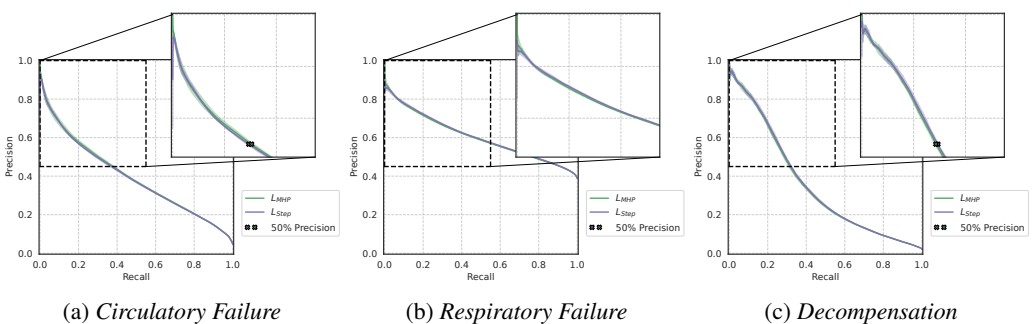

(a) *Circulatory Failure*      (b) *Respiratory Failure*      (c) *Decompensation*

Figure 19: **Precision-recall curves of multi-horizon prediction and temporal label smoothing with** $q_{step}$. Both curves overlap, as suggested by metrics in Table 3, further demonstrating that the multiple outputs of multi-horizon prediction do not lead to superior performance, and supporting assumptions in Proposition 1.

performance of temporal label smoothing combined with a weighted cross-entropy objective is given in Figure 20. Balanced reweighting per class results in a performance drop, as observed when applied to traditional cross-entropy (see Section 5.1, Figure 5). Another possible approach to combine these methods would be to leverage temporal information in sample re-weighting, and we reserve this investigation for further work.

Similarly, no additional performance gains were obtained from combining multi-horizon prediction or focal loss with temporal label smoothing over using TLS with cross-entropy loss.

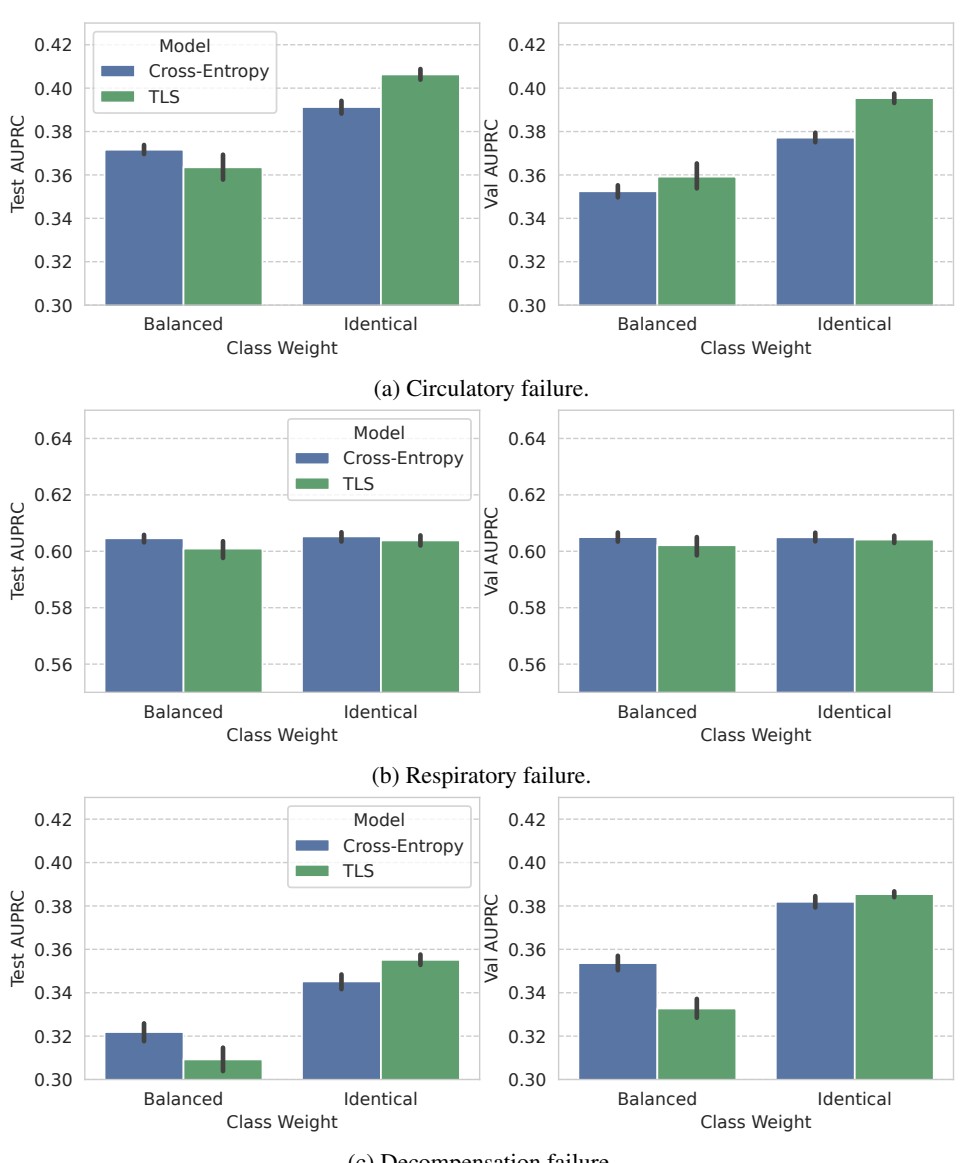

(a) Circulatory failure.

(b) Respiratory failure.

(c) Decompensation failure.

Figure 20: **AUPRC performance of temporal label smoothing combined with weighted cross-entropy**. (Left) Test set performance. (Right) Validation set performance.

