# OpenReview forum: "Temporal Label Smoothing for Early Prediction of Adverse Events"
_ICLR.cc/2023/Conference — Submitted to ICLR 2023_

### Official Review · Reviewer_JLK5 · 2022-10-22

**Confidence:** 4
**Correctness:** 3
**Technical Novelty And Significance:** 2
**Empirical Novelty And Significance:** 2
**Recommendation:** 3

**Clarity, Quality, Novelty And Reproducibility:**

**Clarity.** The paper is mostly clear and easy to follow. Note that I found Figure 2 somewhat difficult to follow (see weakness point #2), and there are some typos that should be fixed (see weakness point #7).

**Quality/novelty.** The paper isn't particularly novel (see weakness point #1), which detracts from the significance of the work. Meanwhile, I have concerns about how much benefit there is, practically speaking, of the proposed method (see weakness point #5). Moreover, I think there aren't enough baselines considered (see weakness point #6).

**Reproducibility.** The authors have provided code. I have not carefully looked at it though.

**Strength And Weaknesses:**

Strengths:
1. The problem being addressed is well-motivated.
2. The proposed TLS method is straightforward to understand.

Weaknesses:
1. The proposed TLS method appears to be a rather incremental advance, extending label smoothing to the temporal setting in a straightforward manner, and then the choice of how the temporal component is parameterized comes off as ad hoc.
2. It took me a bit of time to figure out how to interpret Figure 2. If at all possible, I would suggest providing some additional explanation to make parsing this figure easier.
3. In terms of discussing related work, I think it would be helpful to compare the proposed method with deep imbalanced regression [Yang et al., 2021], which also involves imbalanced data and label smoothing (I realize that the proposed approaches are different and the latter isn't tailored for early adverse event prediction; I think these authors also did look at time series data as one of their examples).
4. In Table 2, I would suggest providing some more significant figures or at least stating something about the p-values that are "0.00" as I would guess they aren't actually exactly equal to 0.
5. Statistical significance aside, I think it would be helpful to provide more commentary on how much of a *practical* difference there is between the proposed temporal label smoothing method and the different baselines. Basically even if there's a statistically significant difference in an evaluation metric, the difference might not be practically significant (some of the numbers from TLS look quite close to those of the multi-horizon baseline, for instance). If it's at all possible to try to quantify the difference in terms of how much earlier TLS can predict an outcome compared to different baselines, that would be helpful (in this case, just as an example, it could be that TLS can in some sense predict as accurately as the multi-horizon baseline 5 seconds faster, and that this 5-second difference is statistically significant but one could argue that 5 seconds is not going to make a difference practically in many clinical applications).
6. I find the number of baselines to be rather small, and it is unclear to me why this is the case. There are *many* machine learning and data mining methods proposed for early prediction/detection/classification of various critical events. I would suggest doing a more thorough literature search to find additional baselines to try. As just a few examples of existing work (this listing is very much non-exhaustive), see the work by Xing et al. [2009], He et al. [2013], Chen et al. [2013], and Lauritsen et al. [2020], or the recent software package by Tavenard et al [2020]; note that some of these would require some sort of conversion of irregularly sampled data into regularly sampled numerical time series but there are straightforward ways to do this (e.g., discretizing time and using some imputation strategy like forward filling to fill in missing values). Moreover, for the specific applications considered, I think it's also worthwhile comparing to any sort of early detection method that is actually currently in clinical use, if any (if not, then it would be helpful indicating that this is the case).
7. There are small typos here and there. Please proofread carefully. As a few examples (not exhaustive), equation (2) ends in "[" which looks like a typo, the third line of the first full paragraph of Section 4.2 has the misspelled word "reweigthing", and the last sentence of the first full paragraph of Section 4.3 has a missing space after a period: ".Thus".

References:
- George H. Chen, Stanislav Nikolov, Devavrat Shah. A Latent Source Model for Nonparametric Time Series Classification. NeurIPS 2013.
- Guoliang He, Yong Duan, Tieyun Qian, Xu Chen. Early prediction on imbalanced multivariate time series. CIKM 2013.
- Simon Meyer Lauritsen, Mads Ellersgaard Kalør, Emil Lund Kongsgaard, Katrine Meyer Lauritsen, Marianne Johansson Jørgensen, Jeppe Lange, Bo Thiesson. Early detection of sepsis utilizing deep learning on electronic health record event sequences. AIIM 2020.
- Romain Tavenard, Johann Faouzi, Gilles Vandewiele, Felix Divo, Guillaume Androz, Chester Holtz, Marie Payne, Roman Yurchak, Marc Rußwurm, Kushal Kolar, Eli Woods. Tslearn, A Machine Learning Toolkit for Time Series Data. JMLR 2020.
- Zhengzheng Xing, Jian Pei, Philip S. Yu. Early Prediction on Time Series: A Nearest Neighbor Approach. IJCAI 2009.
- Yuzhe Yang, Kaiwen Zha, Ying-Cong Chen, Hao Wang, Dina Katabi. Delving into Deep Imbalanced Regression. ICML 2021.

**Summary Of The Paper:**

This paper extends label smoothing [Szegedy et al., 2016] to temporal data, and relates the proposed temporal label smoothing approach to multi-horizon prediction. Experiments on real data show that the proposed temporal label smoothing approach outperforms several baselines.

**Summary Of The Review:**

Overall, I find this paper to be incremental and it's unclear to me how practically significant the improvements are of the proposed method vs the baselines evaluated. Furthermore, I think that there simply aren't enough baselines considered.

---

> ### Author Response · Authors · 2022-11-11
> **Answer to Reviewer JLK5 (1/3)**
>
> We would like to first thank reviewer JLK5 for his comments and for taking the time to review our work. In the following, we answer questions and issues raised and hope to address your concerns.
>
> **Novelty & Contribution**
> > “The proposed TLS method appears to be a rather incremental advance, extending label smoothing to the temporal setting in a straightforward manner, and then the choice of how the temporal component is parameterized comes off as ad hoc.”
>
> We agree with the reviewer's observation that our work is incremental and proposes an extension of label smoothing. However, we rather see this as a strength, as it is built upon existing works and addresses clear limitations of label smoothing for early prediction of events in time series (discussed in Section 3.1), while remaining simple enough to effortlessly integrate into existing frameworks (see implementation in Appendix C.2). We also show a theoretical link to an established training approach for early prediction (multi-horizon prediction) and we achieve superior performance while overcoming the need for additional model complexity.
>
> **Clarity**
> > “It took me a bit of time to figure out how to interpret Figure 2. If at all possible, I would suggest providing some additional explanation to make parsing this figure easier.”
>
> Our goal with Figure 2a was to highlight the degradation in model performance, when trained with cross-entropy, as a function of distance to the event time. These figures highlight the large model confusion near the label boundary (maximum false positive rate and minimum true positive rate) at $t_e - h$, while performance is best closer to the event occurrence ($t_e$) and away from it ($t_e-2h$). We have updated the caption of Figure 2 to clarify this.
>
> **Significant figures for p-values**
> > "In Table 2, I would suggest providing some more significant figures or at least stating something about the p-values that are "0.00" as I would guess they aren't actually exactly equal to 0."
>
> In a paired Student’s t-test, two datasets $D_1$ and $D_2$ are compared by matched pairs of data points. In our case, we pair results obtained from different methods based on the random seed used in training (which impacts model initialization and the order in which training data is seen). If, in each pair, the sample from $D_1$ is always greater than its paired item from $D_2$, then the p-value of the Hypothesis $\mu(D_2) > \mu(D_1)$  is equal to 0, validating the alternative hypothesis $\mu(D_2) < \mu(D_1)$. In some of our experiments in Table 2, we found TLS to outperform MHP for each of the ten training instances, resulting in p-values of 0.0. We hope this clarifies your issue.
>
> We have moved the footnote mark about this (bottom of page 7) to make it more visible to the reader.
>
> **Practical performance difference between TLS and baselines**
> > “Statistical significance aside, I think it would be helpful to provide more commentary on how much of a practical difference there is between the proposed temporal label smoothing method and the different baselines. [...] If it's at all possible to try to quantify the difference in terms of how much earlier TLS can predict an outcome compared to different baselines, that would be helpful”
>
> We agree that the statistical significance of metrics does not necessarily translate into practical usefulness. This is why we explored event-based metrics in addition to timestep levels as in prior works (Hyland et al. 2020, Tomasev et al. 2019) which are closer to the task clinicians perform. To further quantify the practical impact of TLS, we extended the paragraph “Event-based analysis” in Section 5.1. For circulatory failure, this represents 7.4 % more events than our best baseline (multi-horizon prediction): this corresponds to reducing the number of missed events in the test set by a factor of 2, from 303 to 152 out of 2045 events on average. In addition, within these events not captured by MHP, TLS predicts them on average 104 minutes before the event, giving clinicians significant time to take action and avoid patient degradation.
>
> **Presentation**
> > “There are small typos here and there. Please proofread carefully."
>
> In equation 2, our choice of bracket implies that d can be between 0 and $t_e(i,t) - t$ but can not be equal to the latter term as labels are not defined at event time $t_e$. We have changed our notation to make this clearer.
>
> Thank you very much for pointing some typos. We have carefully proofread the manuscript and hope to have eliminated all others.

---

> > ### Author Response · Authors · 2022-11-11
> > **Answer to Reviewer JLK5 (2/3)**
> >
> > **Related Work**
> > > “In terms of discussing related work, I think it would be helpful to compare the proposed method with deep imbalanced regression [Yang et al., 2021], which also involves imbalanced data and label smoothing (I realize that the proposed approaches are different and the latter isn't tailored for early adverse event prediction; I think these authors also did look at time series data as one of their examples).”
> >
> > Thank you for pointing out this interesting work. Deep imbalanced regression (DIR), by Yang et al., 2021, can be applied to time series (authors explore the SHHS dataset) but tackles a regression task, aiming to predict continuous values based on full time series.
> >
> > Unfortunately, this method is not applicable to early event prediction – a classification problem. As detailed in the last paragraph of Section 3.1, their task differs from early event prediction because each time series represents a single sample. Thus, samples can be considered i.i.d. We expand on the distinction between our task and time-series classification in the following point.
> >
> > Finally, the label smoothing proposed by Yang et al., 2021, is applied over the distribution of continuous regression labels using a symmetric kernel from Parzen, 1962. The motivation behind this is to increase the negative Pearson correlation with the test error distribution, not to avoid model over-confidence as in label smoothing.
> >
> > This interesting method could certainly be applied to time-series regression tasks, such as the prediction of remaining length of stays at a specific time point (for instance, 24 hours after admission). This is also a common benchmark task on electronic health records, as detailed in the HIRID and MIMIC benchmark papers: Harutyunyan et al., 2019 and Yeche et al., 2021. However, it is not applicable to the different problem formalism of early prediction of adverse events, a classification task with temporally dependent samples.
> >
> > > “I find the number of baselines to be rather small, and it is unclear to me why this is the case. There are many machine learning and data mining methods proposed for early prediction/detection/classification of various critical events. [...] Moreover, for the specific applications considered, I think it's also worthwhile comparing to any sort of early detection method that is actually currently in clinical use, if any (if not, then it would be helpful to indicate that this is the case).”
> >
> > Thank you for pointing out these interesting works. Unfortunately, we find that these methods are not applicable to our task and could not compare against them as baselines.
> >
> > **Distinction between early prediction of events and early time-series classification.** As a first clarification, we have updated the manuscript to mark a clearer distinction between our task of early prediction of adverse events and that of early time-series classification. The latter has been more extensively explored in the literature, as you pointed out, but does not address the same problem.
> >
> > Considering a time series $\mathbf{X}$ up to timestep $t$, early event prediction is concerned with classifying whether a particular event will occur between $t$ and $t+h$, for a fixed horizon $h$. Predictions are made at each timepoint over the entire time series: as multiple samples arise from the same time series and therefore depend on one another over time, these should not be considered as i.i.d.
> > In contrast, early classification of time series aims to regress the first timepoint $t$ at which a label for the entire time series can be predicted with a desired accuracy. A single prediction is made, as soon as possible, for the entire series – which can be considered an independent sample from the dataset.
> >
> > These tasks are related to a certain extent. Early classification of time series can be framed as early prediction of the event “prediction is possible”, where $h=\infty$, given a separate time-series classifier. As a result, we believe that temporal label smoothing could successfully be applied to the latter task, and look forward to exploring this in further work.
> >
> > On the other hand, early event prediction cannot be translated into a simple early classification problem. Regressing the first time $t$ where an event occurrence within horizon $h$ can be predicted assumes access to an early event predictor. As a result, methods designed for early time-series classification do not translate to our problem setting.
> >
> > We make this distinction in Section 3.1 and include a discussion of this related time-series task in Appendix A.3. Unfortunately, Xing et al. [2009], He et al. [2013], Chen et al. [2013] and the method implemented in Tavenard et al [2020] are all designed for early classification of time series and are therefore not applicable to our problem of interest.

---

> > > ### Author Response · Authors · 2022-11-11
> > > **Answer to Reviewer JLK5 (3/3)**
> > >
> > > **Baselines for early prediction of events.** Our task of interest is of particular interest to monitoring systems, where early alarms for imminent adverse events can help avoid their occurrence. An example application is patient monitoring systems, with the development of rule-based estimators of patient risk in intensive care such as the NEWS score (Smith et al., 2013) – the “early detection method that is actually currently in clinical use”.
> > >
> > > More recent machine learning methods for organ failure prediction (e.g. Hyland et al., 2020) have been proposed, but only offer advances in terms of model architecture and remain trained with a cross-entropy objective. This is also the case in Lauritsen et al., 2020, who propose a deep learning LSTM pipeline trained with cross-entropy. We now cite this work in Section 1 and Table 1, as additional evidence that the most common training objective for our task remains cross-entropy.
> > >
> > > Our method contributes a novel objective for early event prediction, and is thus model-agnostic and could be combined with any architecture improvement. To the best of our knowledge, our closest related work, which both addresses the same task and proposes an alternative training objective, is Tomasev et al., 2019, who implement multi-horizon prediction.
> > >
> > >
> > >
> > >
> > >
> > > Thank you very much for your feedback. We look forward to discussing any remaining questions. We would greatly appreciate it if you would increase your score if we have addressed your concerns.

---

> > > > ### Comment · Reviewer_JLK5 · 2022-11-25
> > > > **my score stays the same**
> > > >
> > > > I thank the authors for their detailed response. However, I am not convinced by their author feedback. Overall I think that the authors have not appropriately discussed existing work, especially in addressing how people actually use existing early time-series classification methods in practice. Because the amount of edits seems substantial to me in terms of what is needed to get the draft to a state that would correspond to a clear "accept", I will stay with my current rating and would suggest that the paper go through another round of rigorous review at a future venue, after the authors have substantially improved the paper.
> > > >
> > > > Some remarks regarding the distinction between early prediction of events and early time-series classification: I disagree with the technical distinction made as I don't think the authors correctly state how these existing early time-series classification methods are used. As the authors stated in their author response "A single prediction is made, as soon as possible, for the entire series – which can be considered an independent sample from the dataset." I don't think people actually, in practice, use existing early time-series classification methods where they only make the prediction once. Just as an example, the paper by Chen et al. [2013] provides theory and applied results where they are saying how the prediction accuracy changes as a function of how much of the time series you observe which means that they are actually doing prediction at different time steps allowing for more of the test time series to be observed (again, they provide both theory for this and applied results; in the applied results, they are intentionally trying to show how one might tune on how early we can confidently make a prediction but to do any such tuning, we actually do need to make predictions with different amounts of the same time series available). To make this more apparent, see Algorithm 1 of Nikolov [2012] (one of the co-authors of the Chen et al paper) that explicitly is re-running the classification at multiple consecutive time steps to see whether the same prediction is made multiple times as a heuristic of when we can be confident enough in a prediction. Moreover, in defining the label for each time series, much of these existing methods are flexible and allow the user to define the labels however they want, and in particular, we could define the label to depend on a time horizon. For example, continuing with the Chen et al approach: their approach could be used with time series of healthcare data where we define a time series to be of the "positive" class (i.e., an adverse event happens) if the adverse event happens within the last $h$ time steps of the time series (note that in practice, any time series is finite in length and how much of the ending of it that we get to see depends on data collection but also could intentionally be truncated to be earlier; for example, if a time series has the adverse event happen within the last $h$ time steps but we intentionally truncate it to no longer have the last $h$ time steps, then we could now define the label for this time series to be from the "negative" class since its last $h$ time steps no longer contains the adverse event). Ultimately, I don't buy that in practice anyone would actually use existing early time-series classification methods in a way where they only have do prediction once per time series. One would use these existing methods in a way where you repeatedly make predictions as you see more of a time series.
> > > >
> > > > Regarding the comment "Deep imbalanced regression (DIR), by Yang et al., 2021, can be applied to time series (authors explore the SHHS dataset) but tackles a regression task, aiming to predict continuous values based on full time series": By a trivial standard result in ML theory, if for the regression task, you take the labels to be 1 or 0 (corresponding to binary classification), then this actually turns many regression methods into classifiers. For example, think about k-NN or kernel regression. When we set the labels to be 0 or 1, then these methods are actually now estimating the probability of a test point being in class 0 or class 1; you can threshold on the probability to obtain a final estimate if you want just a single class to be predicted for the test point rather than the probabilities of each of the classes. I don't think the author feedback makes it clear that they're aware that regression methods can often also be used for classification.
> > > >
> > > > Reference:
> > > >
> > > > Stanislav Nikolov. Trend or no trend: a novel nonparametric method for classifying time series. MIT master's thesis 2012.

---

> > > > > ### Author Response · Authors · 2022-12-09
> > > > > **Answer to reviewer JLK5 (1/3)**
> > > > >
> > > > > We thank reviewer JLK5 for engaging in the discussion and for their detailed feedback.
> > > > > Below we answer point per point to their comment to further show why the early classification of time series and early prediction of events in time series are two distinct problems and the methods the reviewers ask us to compare with are not applicable to the early prediction of events.
> > > > >
> > > > >
> > > > > >  I don't think people actually, in practice, use existing early time-series classification methods where they only make the prediction once
> > > > >
> > > > > >  Ultimately, I don't buy that in practice anyone would actually use existing early time-series classification methods in a way where they only have do prediction once per time series. One would use these existing methods in a way where you repeatedly make predictions as you see more of a time series.
> > > > >
> > > > > We grant that these works tend to measure prediction performance as a function of time, either as an *ablation study* or as an *inductive bias* to determine model readiness. In practice, however, at *inference time*, the goal remains to classify the time-series once (and as early as possible). The label of the time-series does not change over time; only the performance of the prediction model as it accesses more and more information about the series.
> > > > >
> > > > > From these considerations, the early time-series classification framework is not directly applicable to early event prediction, where new labels are obtained (and thus new predictions are needed) *at each time point*.
> > > > >
> > > > > We illustrate the above points below, in response to your comments. We hope this clarifies your concerns and look forward to hearing your thoughts in return.
> > > > >
> > > > > > Just as an example, the paper by Chen et al. [2013] provides theory and applied results where they are saying how the prediction accuracy changes as a function of how much of the time series you observe which means that they are actually doing prediction at different time steps allowing for more of the test time series to be observed.
> > > > >
> > > > > Thank you for pointing out this experiment. In Chen et al [2013], predictions are shown to be made at different time steps over the entire series, but the task does not correspond to making regular predictions of labels *changing throughout the time series* as in early prediction of events. In their introduction, Chen et al [2013] clearly state that this problem is not addressed by their framework: “the case where a single time series can have different labels at different times is beyond the scope of this paper”.
> > > > >
> > > > > The applied results of performance against time are given here as an *ablation study*: Figure 2 (a) shows that a NN classifier and a majority voting classifier close the gap with the oracle model as the size of the observed time-series increases. These results evidence the efficiency of the model in classifying time-series as early as possible, but the goal remains to classify the time-series once. In contrast, early event prediction does not grant the flexibility to choose when to make a prediction.
> > > > >
> > > > > We comment on the theoretical results next.
> > > > >
> > > > > >  To make this more apparent, see Algorithm 1 of Nikolov [2012] (one of the co-authors of the Chen et al paper) that explicitly is re-running the classification at multiple consecutive time steps to see whether the same prediction is made multiple times as a heuristic of when we can be confident enough in a prediction
> > > > >
> > > > > In the theoretical framework of Chen et al. [2013] and Nikolov [2012], model performance against time is also used as a *training device* to measure model confidence.
> > > > > Algorithm 1 from Nikolov [2012] aims at finding the right time point to make a prediction for the entire time-series, such that the model is confident enough. As you pointed out, in this work, authors measure model confidence based on the agreement between successive predictions, determining if these all exceed a threshold $\theta$ for $D_{rep}$ consecutive observed time points.
> > > > >
> > > > > Differences in predictions over time (as the size of the observed time-series increases) are used to determine whether or not to classify the time series. Overall, this still gives a single prediction per time series, the task consisting of *when* to make this prediction.

---

> > > > > > ### Author Response · Authors · 2022-12-09
> > > > > > **Answer to reviewer JLK5 (2/3)**
> > > > > >
> > > > > > This framework also assumes that predictions made after successive observations should agree, because they share a unique label. This cannot be assumed in our case, where adding an observation to a time-series can change its label (e.g. entering the horizon $h$ ahead of an event).
> > > > > >
> > > > > > Next, we explore whether we could phrase our early event prediction task such that it could be trained with early time-series classification methods.
> > > > > >
> > > > > > > Moreover, in defining the label for each time series, much of these existing methods are flexible and allow the user to define the labels however they want, and in particular, we could **define the label to depend on a time horizon**. For example, continuing with the Chen et al approach:  their approach could be used with time series of healthcare data where we define a time series to be of the "positive" class (i.e., an adverse event happens) if the adverse event happens within the last time steps of the time series
> > > > > >
> > > > > > Thank you for your suggestion on how we could frame the early event prediction task as an early classification problem.
> > > > > >
> > > > > > For a time-series of length $T$ with an event occurring between $t_1$ and $t_2$, our task is to predict a positive label between $t_1-h$ and $t_1$, and a negative label before $t_1-h$ and after $t_2$.
> > > > > >
> > > > > > Following the approach taken in Chen et al. [2013],
> > > > > >  - We could truncate our time-series into chunks with positive labels if their last timepoint falls between $t_1-h$ and $t_1$ and negative otherwise (excluding series which end between $t_1$ and $t_2$). This now corresponds to the (non-early) classification task considered by our baselines (all using a cross-entropy training objective), but this does give us the opportunity for explicit comparison to Chen et al. who instead use NN or MV classifiers. Note that the goal is now not to predict *early* (i.e. earlier than the specified clinical horizon $h$), but accurately. Unfortunately, these methods rely on a similarity measure between time series based on temporal shifts – which may introduce label artifacts for our task.
> > > > > > - An alternative is to set a positive label if the last timepoint falls in between $t_1$ and $t_2$, and negative otherwise. This allows us to reframe our task as the closest early classification problem.
> > > > > > Training becomes challenging, as predicting a positive label for a time-series *before the event* would correspond to a *negative example* in the training data (same time-series, with no event in the last time points). An early classification model could therefore not be expected to predict the event much earlier than the event time $t_1$.
> > > > > > Of course, we could try to overcome this with labeling choices, e.g. specifying that negative examples should end at least X time points before an event, or should be taken from patients who do not undergo adverse events over their stay. These modeling considerations are significant modifications to our originally defined task – which likely explains why the early time-series classification framework has not been used in the clinical event prediction literature, with variable labels over individual time-series.
> > > > > >
> > > > > > Note that both approaches do not take the temporal consistency of labels into account, unlike our proposed method. As mentioned in our previous rebuttal, we could however introduce temporal label smoothing into the time-series classification framework (e.g. smoothing labels based on the proximity of the last time-point to the event time) as an extension.

---

> > > > > > > ### Author Response · Authors · 2022-12-09
> > > > > > > **Answer to reviewer JLK5 (3/3)**
> > > > > > >
> > > > > > > > By a trivial standard result in ML theory, if for the regression task, you take the labels to be 1 or 0 (corresponding to binary classification), then this actually turns many regression methods into classifiers.
> > > > > > > For example, think about k-NN or kernel regression. When we set the labels to be 0 or 1, then these methods are actually now estimating the probability of a test point being in class 0 or class 1;
> > > > > > > you can threshold on the probability to obtain a final estimate if you want just a single class to be predicted for the test point rather than the probabilities of each of the classes.
> > > > > > >
> > > > > > >
> > > > > > > We understand that some regression methods can be used for binary classification by treating the labels as 1 or 0, and considering the regression output as some measure of the probability for each class. Still, regression methods adapted for binary classification will not necessarily perform as well as classification methods, specifically designed to predict class labels. As a result, we respectfully argue that using a regression method for binary classification is not necessarily a trivial result in machine learning theory.
> > > > > > >
> > > > > > > In addition, Yang et al. [2021] are explicitly “motivated by the intrinsic difference between categorical and continuous label space” when proposing DIR. Figure 2 illustrates that “the error distribution is very different for [a regression task] with continuous label space, even when the label density distribution is the same as [a classification task]”.
> > > > > > >
> > > > > > > As an error on the regression task is less correlated with the label distribution, authors argue that compensating for imbalance in the empirical label distribution does not necessarily work well in the continuous case. This motivates Yang et al. to first smooth labels over their entire distribution, giving an effective label distribution that correlates well with error distribution. Only then do imbalanced classification techniques help (e.g. re-weighting and focal loss, to which we compare in our work).
> > > > > > >
> > > > > > > Overall, our experimental baselines and discussion include (non-temporal) label smoothing, imbalanced classification techniques (focal loss, weighted cross-entropy), as well as combinations of temporal label smoothing with reweighting techniques (Appendix D.7) – which can be compared to the techniques proposed in Deep Imbalanced Regression. We will add a note to clarify this in our manuscript. We find that our approach outperforms all others, which can be attributed to leveraging temporal dependencies between samples during training.

---

> > > > > > > > ### Comment · Reviewer_JLK5 · 2022-12-11
> > > > > > > > **reply to author reply**
> > > > > > > >
> > > > > > > > I thank the authors for providing a detailed response. I want to emphasize that I only picked on the Chen et al (2013) and Nikolov (2012) papers as examples. There has been many decades of work on early time series classification, early warning systems, and related problems. Nitpicking on the Chen et al or Nikolov papers does not address the broader question that I was trying to get at in my original review (specifically point #6 of my original review under "weaknesses"): to paraphrase, I want to understand to what extent the existing mountain of literature (and methods developed) on early prediction/detection/classification systems apply to the problem being addressed, and whether it is really the case that the number of baselines should be as "thin" as what is considered in the paper draft. Also, I would emphasize that just because a baseline does not *exactly* solve the same problem being addressed does not mean that it cannot be modified in a fairly trivial way to do so.
> > > > > > > >
> > > > > > > > To point out some issues I still find with the most recent author response:
> > > > > > > >
> > > > > > > > - Regarding the statement "Unfortunately, these methods rely on a similarity measure between time series based on temporal shifts – which may introduce label artifacts for our task": I think there's a misunderstanding here. The framework by Chen et al literally allows for the amount of shift to be 0, i.e., for shifts to not be used. From my understanding, this is a hyperparameter setting. Thus, their framework does not require there to be shifts and a special case of their theoretical and applied results would be to put in a constraint that there's no shifting allowed whatsoever. Also, I think it's helpful to think about why they even are suggesting possibly using time shifts: depending on the application, it might make sense to solve a time series alignment problem, and in fact many time series classification papers use much more complicated alignment strategies than simply shifting (e.g., using dynamic time warping would be a classic example). Obviously for many time series analysis problems, shifting in time wouldn't make sense.
> > > > > > > >
> > > > > > > > - Regarding the statement "As a result, we respectfully argue that using a regression method for binary classification is not necessarily a trivial result in machine learning theory": what I had stated in my previous comment is a textbook result at this point. Chapter 1 of the book "A Distribution Free Theory of Nonparametric Regression" (Gyorfi et al 2002) explicitly walks through an introduction of regression followed by how binary classification is a special case (look at sections 1.1 through 1.4), and the fundamental result there (Theorem 1.1) is well-known in statistical machine learning. It relates regression error to classification error (specifically where you learn a regression function to estimate the probability of a data point being in class 0 or class 1; then the classifier just thresholds on the regression function's output). Note that the book by Gyorfi et al is *not* establishing this result as new; it is standard and has been known for such a long time that it is just in the introduction of the book basically as a review.
> > > > > > > >
> > > > > > > > - Regarding the statement "These modeling considerations are significant modifications to our originally defined task – which likely explains why the early time-series classification framework has not been used in the clinical event prediction literature, with variable labels over individual time-series": I think one has to be careful with suggesting a claim regarding why existing early prediction/detection/classification methods aren't actually used in clinical event prediction literature. A large issue here is just the disconnect between methods developers and clinicians: that a method isn't used in clinical practice or in clinical event prediction literature, for instance, might not be because the method doesn't make sense to be used. A different reason is that it could be that the collaboration just didn't get formed between methods developers and clinicians. I'm inclined to believe that there are lots of methods developed, whether early prediction/detection/classification related or otherwise, that would work in the clinical setting but there just hasn't been that partnership formed yet (and maintaining such a partnership/interdisciplinary collaboration could take a lot of work of course).
> > > > > > > >
> > > > > > > > Overall, I still think that this paper would really benefit from a major revision to demonstrate more awareness of existing literature, and to be more careful with the claims it is making.

---

> > > > > > > > > ### Author Response · Authors · 2022-12-12
> > > > > > > > > **Response to Reviewer JLK5 (1/2)**
> > > > > > > > >
> > > > > > > > > Dear Reviewer JLK5,
> > > > > > > > >
> > > > > > > > > Thank you for engaging once again with our response. We clarify some final points before the end of the discussion period.
> > > > > > > > >
> > > > > > > > > > to paraphrase, I want to understand to what extent the existing mountain of literature (and methods developed) on early prediction/detection/classification systems apply to the problem being addressed, and whether it is really the case that the number of baselines should be as "thin" as what is considered in the paper draft. Also, I would emphasize that just because a baseline does not exactly solve the same problem being addressed does not mean that it cannot be modified in a fairly trivial way to do so
> > > > > > > > >
> > > > > > > > > We understand your remaining concern and have repeatedly outlined reasons for which early time-series classification methods are not applicable to our event prediction problem. In our previous answer, we proposed different “trivial modifications” of early classification methods to compare to our framework, with different problem formulations showing either that we do compare against their proposed training objectives (naive classification) or that the problem becomes ill-posed (selecting when to make a prediction).
> > > > > > > > >
> > > > > > > > > For clarity, we restate the key difference between the framework or early time-series classification/regression and early event prediction:
> > > > > > > > > - In early event prediction (the problem we are interested in), the input consists of a time series $[x_{i,1},\ldots, x_{i,T_i}]$ and labels $[y_{i,1},..., y_{i,T_i} ]$ correspond to whether an event will occur in the next $h$ timesteps i.e $y_{i,t} =  1[ \sum_t^{t+h} e_i > 0] $, where $1$ is the indicator function.
> > > > > > > > >
> > > > > > > > > - In early classification of time-series, the input is a partial observation of a time series $[x_{i,1},..., x_{i,K}]$ with $K< T_i$ and a single label exists for the fully-observable time series $y_i$. The goal is to find the smallest $K$ such that $y_i$ is predicted with sufficient accuracy.
> > > > > > > > >
> > > > > > > > > All works on early warning systems, such as Lauritsen et al. 2020 (which you mentioned), as well as Futoma et al. 2017, Hyland et al. 2019, Tomasev et al. 2019, Roy et al. 2021 (discussed in our manuscript) fall in the first category. In all cases, the training objective is a form of cross-entropy, focal, or multi-horizon loss. We compare our method to these four objectives and show that temporal label smoothing matches or outperforms these across all tasks considered.
> > > > > > > > >
> > > > > > > > > All the other works you have cited fall into the second category: Xing et al. 2019, Chen et al. 2013, He et al. 2013., and Tavenard et al. 2020. These works are centered on finding *when* to make a prediction for a given time series, based on a classifier trained by cross-entropy, nearest-neighbor, or majority voting.
> > > > > > > > > At the risk of repeating ourselves, we note two important points:
> > > > > > > > > 1) The assumption of a single label per time series makes these methods inapplicable to the dynamic labels of early event prediction, without significant modification (which we have shown to be non-trivial in our previous answer).
> > > > > > > > > 2) Without the time-step selection component (‘when’ to predict), all proposed methods fall back to naive classification (or regression) training objectives.
> > > > > > > > >
> > > > > > > > > Once again, while we believe that temporal label smoothing could be used to improve performance on early time-series classification, we do not find a trivial modification of this setup that allows us to use these methods for our task.
> > > > > > > > >
> > > > > > > > > > Nitpicking on the Chen et al or Nikolov papers does not address the broader question that I was trying to get at in my original review
> > > > > > > > >
> > > > > > > > > We entirely agree and this is why our original answer outlined the key differences between early event prediction and early classification of time series, in general. Our last answer responded to your comments on specific examples. We believe this gave us the opportunity to demonstrate why the early time-series classification framework cannot be trivially modified to accommodate early event prediction.
> > > > > > > > >
> > > > > > > > > > The framework by Chen et al literally allows for the amount of shift to be 0, i.e., for shifts to not be used. From my understanding, this is a hyperparameter setting. Thus, their framework does not require there to be shifts and a special case of their theoretical and applied results would be to put in a constraint that there's no shifting allowed whatsoever.
> > > > > > > > >
> > > > > > > > > Thank you for pointing this out, we agree. We wanted to note that we could not use the distance metric by Chen et al: as a result, comparing to their work would boil down to comparing to an NN-classifier with Euclidian distance.

---

> > > > > > > > > > ### Author Response · Authors · 2022-12-12
> > > > > > > > > > **Response to Reviewer JLK5 (2/2)**
> > > > > > > > > >
> > > > > > > > > > > Chapter 1 of the book "A Distribution Free Theory of Nonparametric Regression" (Gyorfi et al 2002) explicitly walks through an introduction of regression followed by how binary classification is a special case (look at sections 1.1 through 1.4), and the fundamental result there (Theorem 1.1) is well-known in statistical machine learning.
> > > > > > > > > >
> > > > > > > > > > Thank you for this additional reference, we understand that this is an established theoretical result and should have clarified our statement: this is not necessarily a trivial result in machine learning *practice*. Regression methods adapted for binary classification do not necessarily perform as well as specific binary classification methods. In any case, we do not believe this point to be critical to our comparison of related work. The work that sparked this discussion (Yang et al. 2021) is explicitly “motivated by the intrinsic difference between categorical and continuous label space”. They propose a method to make *imbalanced classification techniques* useful for regression; and we compare to these techniques in our work (re-weighted cross-entropy and focal loss).
> > > > > > > > > >
> > > > > > > > > > > I think one has to be careful with suggesting a claim regarding why existing early prediction/detection/classification methods aren't actually used in clinical event prediction literature. A large issue here is just the disconnect between methods developers and clinicians: that a method isn't used in clinical practice or in clinical event prediction literature, for instance, might not be because the method doesn't make sense to be used. A different reason is that it could be that the collaboration just didn't get formed between methods developers and clinicians. I'm inclined to believe that there are lots of methods developed, whether early prediction/detection/classification related or otherwise, that would work in the clinical setting but there just hasn't been that partnership formed yet (and maintaining such a partnership/interdisciplinary collaboration could take a lot of work of course).
> > > > > > > > > >
> > > > > > > > > > We agree that there are likely many clinical applications suited to the early classification of time series. Our previous paragraphs attempted to convince you that our specific problem setting is not. Overall, possible reasons for why literature has or hasn’t been used for a particular application remain speculation from both our sides, and we have naturally not included any such interpretation in the proposed manuscript.
> > > > > > > > > >
> > > > > > > > > >
> > > > > > > > > > We hope these additional comments will help clarify the position of our work within the related literature.

---

### Official Review · Reviewer_kP8w · 2022-10-25

**Confidence:** 5
**Correctness:** 2
**Technical Novelty And Significance:** 2
**Empirical Novelty And Significance:** 2
**Recommendation:** 3

**Clarity, Quality, Novelty And Reproducibility:**

The paper solves an important problem and proposes a novel extension of classical method to solve it. However, the presentation and clarity of the method needs to be improved. The impact of the solution is also debatable (which arguably could also have been impaired by the presentation).


**Details Of Ethics Concerns:**

No sub-group level analysis has been presented. The proposed schema, if used in practice, can have unintended effects of amplifying algorithmic bias by preferential selection of samples


**Strength And Weaknesses:**

Some of the key strengths of the paper are as follows;

- The authors study a clinically important problem. The proposed mechanism is simple and a natural extension to classical techniques. Furthermore, being a model agnostic regularization scheme, the proposed solution can have a significant impact on medical AI problems.
- The results at a high level is promising across 2 tasks. The authors have also provided some deliberation on when the method may not be satiable
- Figure 5 and the associated analysis is particularly interesting. The insights about the negative weighting of re-weighting samples has the potential to impact future research

The paper may need to address the following aspects

- First, the presentation of the paper makes it less comprehensible and hard to follow. Figures have been presented without properly marking the x axis (e.g Figure 4 - what does each of the ticks around x-axis represent?). The method also lacks clarity. The authors may want to improve the method presentation and use a glossary to help the readers follow the proposed notations.
- A somewhat related criticism around the presentation can be identified in understanding the effect of the label smoothing technique (for instance Figure 2 is missing x-tick annotations as well). It seems from the illustrations that the regularization technique rewards the model to learn the near horizon data points better. If this understanding is correct, the clinical significance of the model can be debatable. For instance, predictions for certain tasks such as decompensation a few time points from actual event is clinical irrelevant as the actionability of such predictions is low.
- This connects to the third criticism around clinical impact of the work - while the authors have claimed that clinically relevant metrics (e.g in Figure 7) is in scope, there is no sub-group level analysis to make sure the model is not unduly decreasing performance across certain sub-groups. The cutoff criteria for AU-PRC evaluation also seems arbitrary - if this was selected under the guidance of clinicians/practicioners, it should be reported.


**Summary Of The Paper:**

The authors studied the problem of improving model predictions of adverse clinical events by using a novel regularization scheme on the target labels. The main novelty lies in adapting the classical Label Smoothing technique to be temporally aware and empirical results have been presented to justify the claim of effectiveness of the method.


**Summary Of The Review:**

Overall, the authors have proposed an interesting and simple solution to handle multi-horizon forecasts, especially for EHR data. While the presented results are interesting, the clinical meaningfulness of the experiments is not well justified. Some other aspects that may need to be addressed are as below

- Section 4.3, the reported method of providing uncertainty around model predictions is arguably measuring the training stability. There are other and arguably important forms of uncertainty that typically plagues EHR data (such as data uncertainty due to selection bias and model uncertainty due to over-specification). It may be useful for the authors to justify what form of uncertainty are they reporting and how to interpret the results for model usage
- Section 5.1, below Table 2, the authors claim that their method is 'statistically superior' - this is a vague term. They may consider reporting on the 'statistical significance' of the results

---

> ### Author Response · Authors · 2022-11-11
> **Answer to reviewer kP8w (1/2)**
>
> We would like to first thank reviewer kP8w for his comments and for taking the time to review our work. Below, we answer questions and issues raised by the reviewer.
>
> **Presentation**
>
> > “Figures have been presented without properly marking the x axis (e.g Figure 4 - what does each of the ticks around x-axis represent?).” “Figure 2 is missing x-tick annotations as well”
>
> Our goal with Figures 2a and 4 was to highlight the degradation in model performance, when trained with cross-entropy, as a function of distance to the event time. These figures highlight the large model confusion near the label boundary (maximum false positive rate and minimum true positive rate) at $t_e - h$, while performance is best closer to the event occurrence ($t_e$) and away from it ($t_e-2h$).
>
>
> In both cases, we did not label all ticks to avoid overloading the x-axis, with labels such as $\{t_e - h, t_e -\frac{11h}{12}, t_e -\frac{10h}{12}, …  t_e\}$. Note that absolute x-ticks would be different for decompensation and circulatory/respiratory failure tasks, which have different prediction horizons ($h=24$ and $12$ hours respectively). This notation allows us to plot performance for all three tasks on the same axis.
>
>
> We ensured that all our figure's axes were labeled and annotated with respect to the horizon of prediction $h$ and the time of the event $t_e$. In addition to defining $t_e$ and $h$ in the caption of the first figure they appear (Figure 2), we repeat this for Figure 4 in the revised manuscript. We also now include information about the size of x-step $\frac{h}{12}$ in the captions.
>
> **Clarity**
>
> > “The method also lacks clarity. The authors may want to improve the method presentation and use a glossary to help the readers follow the proposed notations.”
>
> Thank you for your comment. We have included a glossary at the beginning of Section 3 to summarize our notation and labeling process. We hope this improves clarity and look forward to hearing your feedback on this.
>
> **Effect of TLS on performance over time**
>
> > “It seems from the illustrations that the regularization technique rewards the model to learn the near horizon data points better. If this understanding is correct, the clinical significance of the model can be debatable. For instance, predictions for certain tasks such as decompensation a few time points from actual event is clinical irrelevant as the actionability of such predictions is low.”*
> As you correctly pointed out, temporal label smoothing penalizes overconfidence near the label boundary region at $t_e - h$, as well as low confidence predictions close to the event at $t_e$. As mentioned in Sections 1 and 3.1, this is motivated by the fact that the signal strength is reduced (or ‘decayed’, as further away from the event) at the label boundary.
>
> If all time points were recalled by the model, we agree that this could lead to predicting events later than regular cross-entropy. However, as shown in Figure 2(a) and Figure 4, for these challenging, noisy tasks, this is not the case. As a result, we observe a much greater timestep recall as we get closer to the event, with no loss of recall at the label boundary for both circulatory failure and decompensation (Figures 7 and 15).
>
>
> This translates directly to an improvement in event recall. To ensure that events are not just recalled later, we plot event recall for different horizons from $0$ hours to $h$ hours in Figures 6(b) and 13. We find that performance is always better or comparable to MHP or cross-entropy. As improvements are observed beyond the first bin (4h for decompensation and 2h for circulatory failure), TLS also improves event recall at horizons where actions can still be taken by clinicians to avoid patient degradation.
>
>
> We also included some more practical details about the improvement in event detection in Section 5.1. For circulatory failure, TLS detects 7.4 % more events than our best baseline (multi-horizon prediction): this corresponds to reducing the number of missed events in the test set by a factor of 2, from 303 to 152 out of 2045 events on average. In addition, within these events not captured by MHP, TLS predicts them on average 104 minutes before the event, giving clinicians significant time to take action.

---

> > ### Author Response · Authors · 2022-11-11
> > **Answer to reviewer kP8w (2/2)**
> >
> > **Subgroup analysis**
> >
> > > “This connects to the third criticism around clinical impact of the work - while the authors have claimed that clinically relevant metrics (e.g in Figure 7) is in scope, there is no sub-group level analysis to make sure the model is not unduly decreasing performance across certain sub-groups.”
> >
> > > “No sub-group level analysis has been presented. The proposed schema, if used in practice, can have unintended effects of amplifying algorithmic bias by preferential selection of samples”
> >
> >
> > We agree that due to the high level of heterogeneity in the data, deep learning model approach for clinical prediction task are prone to be algorithmic bias. To ensure the population-wide improvements of TLS are not achieved by disproportionally favouring specific cohorts, we now provide a sub-group analysis in Appendix D.3. We observe that the improvements observed with TLS are consistent across different age and gender groups.
> >
> > **Clinical significance of results**
> > >“the clinical meaningfulness of the experiments is not well justified”
> >
> > We hope that our above paragraph on model performance over time addresses your concerns about the clinical meaningfulness of our proposed method. We look forward to discussing any remaining concerns.
> >
> > >“The cutoff criteria for AU-PRC evaluation also seems arbitrary - if this was selected under the guidance of clinicians/practicioners, it should be reported.”
> >
> > As for the precision threshold chosen to report recall results, this choice is inspired by recent work on early prediction of adverse clinical events (Hyland et al.,2020, Tomasev et al., 2019, Lauritsen et al., 2020). These prior works, which motivated the development of our method, stress the importance of low false-alarm rates for model acceptance in clinical practice, often favoring a >50% precision threshold (fewer than one false alarm for every predicted event).
> >
> > Note that our AUPRC analysis (both absolute numbers given in Table 2 and full curves in Figs 6a and 14) deliberately considers the full precision range to not focus merely on this aforementioned clinically-relevant region. We find that TLS performance is consistently higher than (or comparable to) other training objectives across the full high-precision region. In fact, the gap increases as we increase the precision threshold. We ensured to make this point clearer in Section 5.1 of the revised manuscript.
> >
> > **Performance uncertainty**
> > > “Section 4.3, the reported method of providing uncertainty around model predictions is arguably measuring the training stability. There are other and arguably important forms of uncertainty that typically plagues EHR data (such as data uncertainty due to selection bias and model uncertainty due to over-specification). It may be useful for the authors to justify what form of uncertainty are they reporting and how to interpret the results for model usage”
> >
> > As you noted, our method is model agnostic. Architecture choice, feature selection, and imputation were adopted from the HiRID and MIMIC benchmark papers to allow for a reliable comparison of methods proposing different training objectives, such as ours. Our goal was to determine whether our training objective could improve performance, when combined with these established models.
> >
> > As mentioned in Appendix C, in contrast to other previous work in the field (Tomasev et al. 2019, Roy et al. 2021), we did not use a pivot bootstrap on the test to compute uncertainties, for reasons highlighted below. This approach accounts for the variation in performance across patients, and should thus account for some selection bias or over-specification. With the uncertainty evaluation framework from Tomasev et al. 2019, bootstrapping patients from our test set 200 times for each training instance, we obtained similar means to Table 2 but with confidence intervals all smaller than or equal to 0.1%. Variance within bootstrap samples from the same training instance is therefore much smaller than across instances. Our alternative uncertainty estimation approach, measuring variability between training runs, returns more conservative estimates, and was thus chosen for all results reported in our work.
> > We now include this discussion in Appendix D.4 for the reader.
> >
> > **Statistical significance of results**
> >
> > > “Section 5.1, below Table 2, the authors claim that their method is 'statistically superior' - this is a vague term. They may consider reporting on the 'statistical significance' of the results”
> >
> > Thank you for pointing this out. We have reformulated this sentence in Section 5.1 to make this clearer to the reader, and hope this addresses your concern.
> >
> > Thank you very much for your feedback. We look forward to discussing any remaining questions. We would greatly appreciate it if you would increase your score if we have addressed your concerns.

---

### Official Review · Reviewer_zja8 · 2022-10-25

**Confidence:** 4
**Correctness:** 3
**Technical Novelty And Significance:** 2
**Empirical Novelty And Significance:** 2
**Recommendation:** 5

**Clarity, Quality, Novelty And Reproducibility:**

This is a nicely written paper. The notaions used in this paper is clear and the reference to the appendix etc present. Occational, the notaion shows up before first defination like \alpha^{exp} show in Figure 3a before defining in equaiton (5), Table 1 refers to A3 which is clear while still may not make too much sense to reader who're not so familar with this topic first time.

The idea of smoothing at the boundary is old in general yet new for this specific topic. The implementation of this smoothing did help improve the performance in two of the three adopted datasets, which are with less repeated AEs.

The smoothing agothrim provided and the adopted GRU/transformer combination looks replicable but I haven't tried to reproduce.

**Details Of Ethics Concerns:**

Found this under review paper with author names listed under
https://arxiv.org/abs/2208.13764
https://deepai.org/publication/temporal-label-smoothing-for-early-prediction-of-adverse-events

**Strength And Weaknesses:**

This paper focuses on the existing practical issue of adverse event (AE) prediciton to propose temporal label smoothing. Detailed backgroud review and ample exploration of performance under multiple existing prediction methods and smoothing methods.

It's nice that the author compared not only AUPRC but also Recall.

This reviewer is wondering how the repeated AEs were presetned in the HiRID dataset and did the author pre-process the data or train the data to add a specific feature indicating first/repeated/worsen/relieved/... AE, and if such effort can help improve the performace, especially for the with frequent repeated AEs scenario, under which the proposed method isn't significant compared to existing methods.


**Summary Of The Paper:**

Motivated to reduce model confidence with stronger smoothing at the class boundary, this paper proposed to Temporal Label Smoothing, for early prediciton of adverse event. Simple form comparison with existing methods and experiments using three ICU datasets are presented. It is also proved that stepwise temporal label smoothing is equivalent to Multi-horizon prediciton.

**Summary Of The Review:**

This paper proposes a smoothed version of regulaization called temporal label smoothing to improve boundary performace. The idea is nice for clinic practise and shown to be with better performace with certain clinic datasets with AEs repeatation rate not so high.

The presentation and idea is nice, though the final performance can be improved.

---

> ### Author Response · Authors · 2022-11-11
> **Answer to reviewer zja8**
>
> We would like to first thank reviewer zja8 for his insightful comments and for taking the time to review our work. Below, we answer questions and issues raised by the reviewer.
>
> **Definition of event labels**
>
> > “This reviewer is wondering how the repeated AEs were presetned in the HiRID dataset and did the author pre-process the data or train the data to add a specific feature indicating first/repeated/worsen/relieved/... AE, and if such effort can help improve the performace, especially for the with frequent repeated AEs scenario, under which the proposed method isn't significant compared to existing methods.”
>
> We would like to clarify that all labels were defined by previous works, as mentioned at the beginning of  Section 4.1, and used as such to ensure a fair and reproducible comparison. Both HiRID and MIMIC benchmarks provide binary event labels, indicating whether the patient is undergoing an event or not based on clinically defined criteria.
> Performance could certainly be further improved with additional annotation of data labels, as you suggested, and by adapting our method to take these into account when smoothing labels.
>
> Still, to ensure that our experiments measured the added value of our smoothing labels over time (and not of additional labelling information), we decided to focus on the problem setting with no annotations. This also results in a fair comparison to studied baselines which access the same label information.
>
> We agree that an exciting extension of TLS could consider higher-dimensional classification problems, but leave this for further work as it requires a new problem formalism and methodology.
>
> **Notation**
>
> > “Occational, the notaion shows up before first defination like $\alpha^{exp}$ show in Figure 3a before defining in equaiton (5), Table 1 refers to A3 which is clear while still may not make too much sense to reader who're not so familar with this topic first time.”
>
> We moved Figure 3 to the bottom of page 4 (instead of the top), to ensure it appears after equation (5) and have included a glossary in Section 3 to clarify our notation. We added a reference in Table 1 to the relevant appendix. We hope these corrections address your concern and look forward to hearing from you if anything remains unclear.
>
> **Performance**
>
> > “The presentation and idea is nice, though the final performance can be improved.”
>
> We agree that the performance of our method is comparable to that obtained with other training objectives for early event prediction in one of the three medical tasks considered.
> Overall, our method outperforms all baselines in all considered metrics for tasks showing a strongly reduced recall with time-to-event (decompensation and circulatory failure, as illustrated in Figure 4). Still, we show that for a task with a close-to-constant performance over time, such as respiratory failure (see Figure 4), our method does not perform worse, as conjectured at the end of Section 4.1. In fact, our method achieves superior performance along metrics chosen to be clinically relevant (recall close to event occurrence, and event recall – as discussed in Section 5.1 and 5.3).
>
>
> In practice, its improved AUPRC allows our method to accurately predict a greater number of adverse events (Figures 6b and 13). For circulatory failure, this represents 7.4 % more events than our best baseline (multi-horizon prediction): this corresponds to reducing the number of missed events in the test set by a factor of 2, from 303 to 152 out of 2045 events on average. In addition, within these events not captured by MHP, TLS predicts them on average 104 minutes before the event, giving clinicians significant time to take action and avoid patient degradation.
>
> We have added this analysis in Section 5.2, and hope it further convinces you of the added value of temporal label smoothing in improving performance in early prediction tasks.
>
> Thank you very much for your feedback. We look forward to discussing any remaining questions. We would greatly appreciate it if you would increase your score if we have addressed your concerns.

---

### Author Response · Authors · 2022-11-11
**Response to Reviewers and Revised Manuscript**

Dear Reviewers,

Thank you very much for taking the time to read our paper and for giving us your valuable feedback and questions.

The consensus among the reviewers is that our work is well motivated, both in terms of the importance of clinical applications of interest (zja8, kP8w, JLK5). Modular and flexible (kP8w), our method has shown to be effective (zja8, kP8w) and should be useful in a range of applications in the clinical setting (kP8w).

Positive feedback also included comments on the theoretical analysis (zja8) and experimental design (zja8, kP8w), the insightful discussion of the trade-offs and limitations of our approach (kP8w), and the simplicity of our method (JLK5).

We have addressed all questions and comments individually and are looking forward to hearing the reviewers’ thoughts in follow-up. We incorporated the proposed changes and improvements, temporarily highlighted in blue, in a revised version of our paper. Major changes and additions are summarized below:
- To **improve clarity**, we have updated the captions for Figures 2 and 4. We have included a glossary at the beginning of Section 3, detailing our notation and illustrating our early event prediction task.
- We contrast our task of interest to the **literature on early time-series classification**, which addresses a related but distinct problem, in Appendix A.3. We link to this discussion in Section 2 of the main paper. This explains the relatively small number of baselines for our task, largely unexplored in the machine learning literature.
- In Appendix D.3, we include a **performance analysis across patient cohorts** of different ages and gender, to ensure that temporal label smoothing does not amplify unwanted algorithmic biases.
- In Section 5, we clarify the objective of our null-hypothesis and paired t-tests between performance results, to verify the **superior performance of TLS is statistically significant**. We explain how we obtain $p$-values of 0.0 obtained in some cases.
- In Section 5, we also include some numerical results giving an indication for the **practical performance improvements** of TLS: how many more events can we predict with our objective, and how much earlier are these detected?
- Finally, in Appendix C and D.4, we highlight the limitations of estimating performance uncertainty based on bootstraps from the test sets and justify our reported **uncertainty estimation method**, based on variability between training instances.

We again thank all reviewers for their suggestions and very much look forward to hearing their thoughts on our clarifications.

---

### Author Response · Authors · 2022-11-17
**Follow-up on reviews**

Dear Reviewers,

Thank you again very much for taking the time to read and review our paper. We have taken your helpful feedback into account and have carefully answered your questions. We look forward to hearing your thoughts in follow-up and we hope you will increase your scores if we have addressed your concerns.

Could you please let us know if you have further questions? The deadline for manuscript revisions is tomorrow. Thank you for your help!

---

### Author Response · Authors · 2022-11-25
**Follow-up on reviews**

Dear Reviewers,

As the end of the discussion is approaching and we still haven't heard back from any of you, we would like to kindly remind you that we have taken your helpful feedback into account and have carefully answered your questions. We look forward to hearing your thoughts and we hope you will increase your scores if we have addressed your concerns.

We will be happy to answer any remaining questions. Thank you for your help!

---

### Author Response · Authors · 2022-12-12
**General response after discussion period**

Dear Reviewers, Area Chair, and general readers,

We would like to thank again reviewers for taking the time to read and review our manuscript. [Below](https://openreview.net/forum?id=miyZxvBxdoP&noteId=6koIgJJlba) we summarise the positive feedback on our work. We have addressed their main criticisms in our responses and in our updated manuscript.

Unfortunately, we remain disappointed by the lack of engagement from reviewers and the area chair. Two reviewers did not acknowledge nor respond to our rebuttal. We reached out to the area chair both to ask reviewers to engage with our answers, and also to point out the breach of anonymity from Reviewer zja8 in including a link to a possible arXiv version of this work. As we noted in our message to the area chair, publication to archival repositories is explicitly allowed by [ICLR author guidelines](https://iclr.cc/Conferences/2023/CallForPapers).

We would like to additionally thank Reviewer JLK5 for engaging in a detailed discussion in follow-up to our rebuttal, which we believe allowed us to clarify some points. After addressing some of their original criticism, we understand that their remaining reservations are centered around the difference between the task of classifying time series as early as possible (indeed largely studied in the machine learning literature) and our task of predicting the future occurrence of an event within a pre-defined time horizon. In our answers, we have discussed multiple distinctions between these frameworks, and have studied possible reformulations of our problem to allow for comparison: our conclusion is that early time-series classification methods remain inapplicable to our task.


Paper861 Authors

---

### Decision · Program_Chairs · 2023-01-20

**Decision:**

Reject

**Justification For Why Not Higher Score:**

This paper is clearly below the threshold of acceptance.
See the main part of the summary for the list of main issues.

**Justification For Why Not Lower Score:**

n/a

**Metareview: Summary, Strengths And Weaknesses:**

This submission has been thoroughly reviewed by three knowledgeable reviewers. All of them assessed it below the threshold for acceptance. The key limitations brough up include limited novelty (rather straightforward extension of label smoothing to temporal domain), unclear clinical relevance (the main proposed application is to clinical data), some issues with the clarity of presentation of the method, and a narrow scope of experimental comparisons against baselines.